# Approximate Domain Unlearning for Vision-Language Models

**Kodai Kawamura**[* 1,2], **Yuta Goto**[* 1], **Rintaro Yanagi**[3], **Hirokatsu Kataoka**[3,4], **Go Irie**[1]

[1]Tokyo University of Science
[2]National University of Singapore
[3]National Institute of Advanced Industrial Science and Technology (AIST)
[4]University of Oxford

## Abstract

Pre-trained Vision-Language Models (VLMs) exhibit strong generalization capabilities, enabling them to recognize a wide range of objects across diverse domains without additional training. However, they often retain irrelevant information beyond the requirements of specific target downstream tasks, raising concerns about computational efficiency and potential information leakage. This has motivated growing interest in approximate unlearning, which aims to selectively remove unnecessary knowledge while preserving overall model performance. Existing approaches to approximate unlearning have primarily focused on *class unlearning*, where a VLM is retrained to fail to recognize specified object classes while maintaining accuracy for others. However, merely forgetting object classes is often insufficient in practical applications. For instance, an autonomous driving system should accurately recognize *real* cars, while avoiding misrecognition of *illustrated* cars depicted in roadside advertisements as *real* cars, which could be hazardous. In this paper, we introduce *Approximate Domain Unlearning (ADU)*, a novel problem setting that requires reducing recognition accuracy for images from specified domains (e.g., *illustration*) while preserving accuracy for other domains (e.g., *real*). ADU presents new technical challenges: due to the strong domain generalization capability of pre-trained VLMs, domain distributions are highly entangled in the feature space, making naive approaches based on penalizing target domains ineffective. To tackle this limitation, we propose a novel approach that explicitly disentangles domain distributions and adaptively captures instance-specific domain information. Extensive experiments on four multi-domain benchmark datasets demonstrate that our approach significantly outperforms strong baselines built upon state-of-the-art VLM tuning techniques, paving the way for practical and fine-grained unlearning in VLMs. Code : `https://kodaikawamura.github.io/Domain_Unlearning/`.

## 1 Introduction

Pre-trained Vision-Language Models (VLMs) exhibit remarkable generalization capabilities, enabling them to recognize a wide range of object classes without any additional training [Radford et al., 2021, Jia et al., 2021, Li et al., 2022, 2023, Yu et al., 2022]. However, many practical downstream tasks do not require leveraging their full generalization capability. For instance, an autonomous driving system must recognize "cars" and "pedestrians" but does not need to identify "foods" or "groceries". Retaining unnecessary knowledge introduces serious risks, including excessive computational resource

---

[*]Equal contribution.

consumption and potential information leakage [Fredrikson et al., 2015, Bommasani et al., 2021, Shokri et al., 2017]. To address these concerns, approximate unlearning (also known as selective forgetting), which aims to make classification models forget specified knowledge while preserving the rest, has gained significant attention [Shibata et al., 2021, Kuwana et al., 2024, Graves et al., 2021, Ye et al., 2022, Tarun et al., 2023, Fan et al., 2024, Chen et al., 2023, Huang et al., 2024b].

Previous studies on approximate unlearning for pre-trained VLMs have primarily focused on *class unlearning*, which aims to reduce the recognition accuracy of specified object classes while preserving that of the others [Huang et al., 2024b, Kuwana et al., 2024]. A common approach involves increasing the classification loss for the classes to be forgotten (thus reducing their accuracy) while decreasing it for the classes to be retained (thus improving their accuracy). Building on this idea, several methods have been developed, including a technique that optimizes the gradient direction for more effective unlearning [Huang et al., 2024b] and an approach tailored for black-box VLMs [Kuwana et al., 2024].

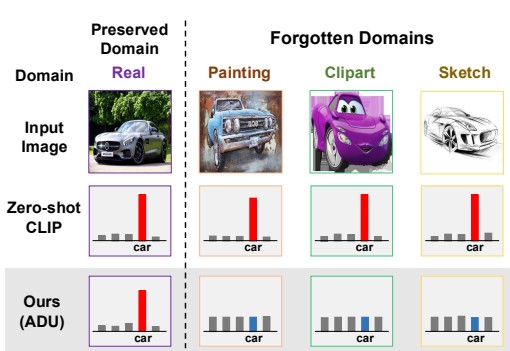

Figure 1: **Illustration of Approximate Domain Unlearning (ADU).** ADU is a novel approximate unlearning problem introduced in this paper. Unlike existing approximate class unlearning tasks, ADU requires retraining a pre-trained Vision-Language Model (VLM) so that it cannot recognize images from specified domains (*painting*, *clipart*, *sketch* in the figure) while preserving its ability to recognize images from other domains (*real* in the figure).

However, we argue that merely forgetting object classes is often insufficient for real-world applications. Recall the example of an autonomous driving system. Such a system must accurately recognize *real* cars to control following distances and prevent collisions. In contrast, if an advertisement depicting *illustrated* cars on the roadside is mistakenly recognized as *real* cars, undesirable behaviors could be triggered, potentially compromising safety. Indeed, due to their strong domain generalization capability, pre-trained VLMs can indiscriminately recognize both *real* and *illustrated* cars as "cars." In such cases, simply forgetting the object class "car" is inadequate; a more fine-grained selective forgetting technique is required to preserve the recognition accuracy of *real* cars while reducing it for *illustrated* cars.

In this paper, we introduce a novel variant of the approximate unlearning problem, called *Approximate Domain Unlearning (ADU)*. Fig. 1 provides an illustration of the proposed ADU task. While conventional approximate class unlearning aims to retrain a pre-trained VLM so that it cannot recognize specified object classes, ADU requires tuning the model so that it cannot recognize images from specified domains while preserving its recognition ability for other domains. For example, a pre-trained VLM may initially recognize a car regardless of its domain, whether it appears in a *real* photograph, *painting*, *clipart*, or *sketch*. If the *real* domain is designated to be retained, ADU selectively untrains the model so that only *real* images of cars remain recognizable, while recognition accuracy for cars in other domains is reduced. ADU is fundamentally different from the traditional class unlearning setting and has not been explored in prior research. As discussed above, it opens a new direction in approximate unlearning, motivated by practical considerations.

ADU also introduces a new technical challenge. A straightforward idea to address ADU would be to adapt the common approximate class unlearning strategy—minimizing the classification loss for domains to be retained while maximizing it for those to be forgotten [Huang et al., 2024b, Kuwana et al., 2024]. However, this approach fails to deliver satisfactory domain unlearning performance. This limitation arises from the strong domain generalization capability of pre-trained VLMs, leading to significant entanglement between different domain distributions in the latent feature space [Kumar et al., 2022, Wang et al., 2024], unlike class distributions, which are typically well-separated. Consequently, attempts to preserve or degrade recognition accuracy for a specific domain inevitably affect all domains, making domain-wise control over feature representations inherently difficult. To address this issue, we propose a novel approach specifically tailored to the problem of ADU. Specifically, we introduce the *Domain Disentangling Loss (DDL)*, which explicitly encourages separation of domain distributions in the latent space. Furthermore, recognizing that domain characteristics may vary in

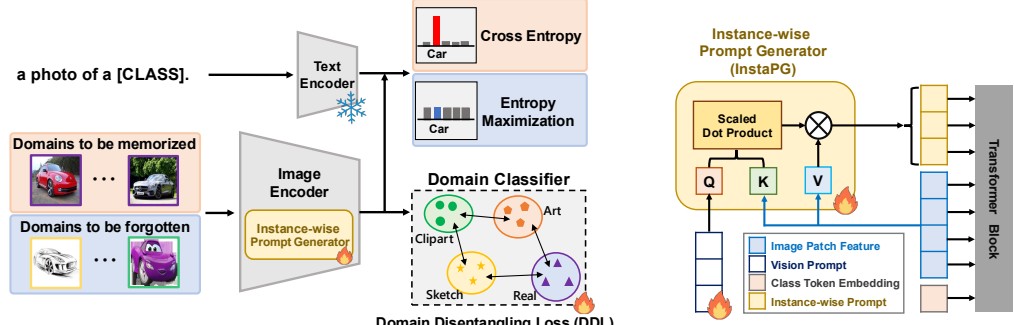

|  |  |
|:---:|:---:|
| (a) Overview of Our Method | (b) Instance-wise Prompt Generator (InstaPG) |

Figure 2: **Overview of Proposed Method.** (a) The common approach to approximate unlearning is to minimize the cross-entropy to the ground truth class labels for the domains to be memorized and to maximize the entropy for the domains to be forgotten. This approach alone is not satisfactory, due to the strong generalization performance of pre-trained VLMs. We therefore introduce two techniques to facilitate ADU; Domain Disentangling Loss (DDL) to disentangle the domain distrubtions in the latent feature space and Instance-wise Prompt Generator (InstaPG) to capture image-level differences of domains. (b) InstaPG utilizes an attention mechanism where the vision prompt acts as the query and the image patch features serve as the key and value. Through this mechanism, instance-wise prompts are dynamically generated, allowing the model to adaptively refine prompts based on individual image characteristics.

strength and spatial extent across images, we introduce an *Instance-wise Prompt Generator (InstaPG)* to adaptively model these properties on a per-image basis. Experiments on four multi-domain image benchmark datasets demonstrate that our method significantly outperforms strong baselines built upon state-of-the-art VLM tuning techniques, achieving superior domain unlearning performance.

The main contributions of this paper are summarized as follows:

- We introduce *Approximate Domain Unlearning (ADU)*, a novel problem setting that extends the notion of approximate unlearning to the domain level, addressing practical limitations of conventional class unlearning.

- We propose a novel approach to ADU, featuring the *Domain Disentangling Loss (DDL)* to explicitly disentangle domain distributions in the latent space and the *Instance-wise Prompt Generator (InstaPG)* to adaptively model instance-level domain variations.

- Extensive experiments on four multi-domain image benchmark datasets validate the effectiveness of our approach, showing substantial improvements over strong baselines built upon state-of-the-art VLM tuning techniques.

## 2 Related Work

### 2.1 Machine Unlearning

Machine unlearning aims to remove the influence of specific samples from pre-trained models, motivated by the growing need to eliminate traces of particular data from such models [Shaik et al., 2024, Xu et al., 2024, Cao and Yang, 2015, Neel et al., 2021, Sekhari et al., 2021, Ullah et al., 2021, Guo et al., 2020, Golatkar et al., 2021, Chen et al., 2019, Brophy and Lowd, 2021, Sun et al., 2023]. Existing methods are broadly categorized into two approaches: exact unlearning and approximate unlearning. Exact unlearning either retrains a model from scratch after removing the target samples, or alternatively modifies the model so that its parameter distribution exactly matches that of a model trained from scratch without those samples [Thudi et al., 2022a,b]. While exact unlearning has been extensively studied in the context of convex optimization of linear machine learning models [Guo et al., 2020, Izzo et al., 2021, Neel et al., 2021], applying it to deep neural networks typically requires full retraining [Bourtoule et al., 2021], a process that is computationally prohibitive for practical applications. To address this limitation, recent research has increasingly focused on approximate unlearning [Fan et al., 2024, Ginart et al., 2019, Golatkar et al., 2020, Guo et al., 2020, Jia et al., 2023,

Warnecke et al., 2023, Graves et al., 2021, Chundawat et al., 2023, Huang et al., 2024b, Kuwana et al., 2024, Shibata et al., 2021], which seeks to efficiently approximate the result of exact unlearning without incurring the full computational cost. Huang et al. [2024b] proposed an update rule based on information-geometric gradient directions, which mitigates interference with the classification accuracy of retained classes. Similarly, Kuwana et al. [2024] introduced a black-box unlearning method for VLMs, enabling forgetting in proprietary models with undisclosed internal architectures.

Despite these advances, existing approximate unlearning approaches have focused exclusively on class-level forgetting. To the best of our knowledge, domain-level unlearning has not been explored. This study presents the first investigation of *Approximate Domain Unlearning (ADU)*, introducing a novel framework for selectively unlearning domain-specific information while preserving generalization capability.

## 2.2 Domain Adaptation / Generalization

Domain Adaptation (DA) and Domain Generalization (DG) are two prominent strategies for tackling domain shifts in machine learning. DA assumes access to (labeled or unlabeled) target-domain data during training, while DG requires models to generalize to unseen domains without exposure to target data [Patel et al., 2015, Zhou et al., 2022a]. Recent efforts in these areas have explored various strategies, including adversarial training [Ganin et al., 2016], domain-invariant representation learning [Li et al., 2018b], meta-learning [Li et al., 2018a], and prompt tuning for large-scale pre-trained models [Zhou et al., 2022b, Shu et al., 2022]. A classical and widely adopted technique for distribution alignment in DA and DG is Maximum Mean Discrepancy (MMD) [Gretton et al., 2012], a non-parametric measure that estimates the distance between probability distributions in a Reproducing Kernel Hilbert Space (RKHS). Many domain adaptation methods incorporate MMD as a regularizer to minimize the distance between marginal or joint feature distributions of source and target domains. Representative examples include Deep Adaptation Network (DAN) [Long et al., 2015] and Joint Adaptation Network (JAN) [Long et al., 2017], as well as follow-up works that further refine the alignment process [Yan et al., 2017, Wang et al., 2020, Mekhazni et al., 2020].

Unlike these methods, which *minimize* MMD to enforce domain invariance, our approach *maximizes* MMD within a Domain Disentangling Loss (DDL) to deliberately separate domain-specific representations. This inversion of the conventional alignment objective reflects the fundamentally different nature of ADU, which requires not domain invariance but *domain disentanglement* for the selective removal of unwanted domain-specific knowledge.

# 3 Method

## 3.1 Approximate Domain Unlearning (ADU)

Given a set of training data $\{(\mathbf{x}, y, d)\}$, where $\mathbf{x} \in \mathcal{X}$ represents an input image, $y \in \mathcal{C}$ is the class label, and $d \in \mathcal{D}$ is the domain label, with $\mathcal{X}$, $\mathcal{C}$ and $\mathcal{D}$ denoting the input space, the set of all classes, and the set of all domains, respectively. We define $\mathcal{D}_{\text{memorize}} \subset \mathcal{D}$ as the set of domains to be preserved and $\mathcal{D}_{\text{forget}} = \mathcal{D} \setminus \mathcal{D}_{\text{memorize}}$ as the set of domains to be forgotten. Our goal is to retrain a pre-trained VLM $f$ to maintain the classification accuracy for $\{(\mathbf{x}, y, d)|d \in \mathcal{D}_{\text{memorize}}\}$, while reducing it for $\{(\mathbf{x}, y, d)|d \in \mathcal{D}_{\text{forget}}\}$.

## 3.2 Applying Common Approximate Class Unlearning Approach to ADU

The common approach to the conventional approximate class unlearning is to use two different loss functions; one for retaining the classification accuracy for the classes to be retained and the other for reducing the accuracy for those to be forgotten [Huang et al., 2024b, Kuwana et al., 2024]. Specifically, given a mini-batch $\mathcal{B} = \{(\mathbf{x}_i, y_i, d_i)\}_{i=1}^{|\mathcal{B}|}$, the cross-entropy to the ground truth class labels $\mathcal{L}_{\text{memorize}}$ is minimized for the classes to be retained, and the cross-entropy to the uniform class labels $\mathcal{L}_{\text{forget}}$ is minimized (which corresponds to entropy maximization) for the classes to be forgotten [Kuwana et al., 2024]:

$$\mathcal{L}_{\text{memorize}}(\mathcal{B}) = -\frac{1}{|\mathcal{B}|} \sum_{i=1}^{|\mathcal{B}|} \sum_{j=1}^{|\mathcal{C}|} y_{ij} \log p_{ij}, \tag{1}$$

$$\mathcal{L}_{\text{forget}}(\mathcal{B}) = -\frac{1}{|\mathcal{B}|} \sum_{i=1}^{|\mathcal{B}|} \sum_{j=1}^{|\mathcal{C}|} \frac{1}{|\mathcal{C}|} \log p_{ij}, \tag{2}$$

where $\mathbf{p}_i = (p_{i1}, p_{i2}, \ldots, p_{i|\mathcal{C}|})^\top$ represents the confidence scores of a sample $\mathbf{x}_i$ output by the model, and $\mathbf{y}_i = (y_{i1}, y_{i2}, \ldots, y_{i|\mathcal{C}|})^\top$ denotes the one-hot encoding of the class label $y_i$.

Given this idea, a straightforward approach to ADU would be to adapt these two loss functions, that is, minimizing $\mathcal{L}_{\text{memorize}}$ for $\{(x, y, d) | d \in \mathcal{D}_{\text{memorize}}\}$ and $\mathcal{L}_{\text{forget}}$ for $\{(x, y, d) | d \in \mathcal{D}_{\text{forget}}\}$. However, as we will show later in our experiments, this straightforward approach alone is insufficient to achieve satisfactory ADU performance. This is primarily due to the strong domain generalization capability of pre-trained VLMs. As evidenced by their robustness to domain shifts, the latent space of VLMs highly aligns data distributions across different domains, meaning that covariate shifts are minimal. Consequently, the feature distributions across different domains are highly entangled, making it difficult to effectively control memorization and forgetting on per-domain basis.

### 3.3 Domain Disentangling Loss (DDL)

To address this issue, we propose Domain Disentangling Loss (DDL), which aims to explicitly disentangle the feature distributions among different domains in the latent feature space. The core idea is that if the feature distributions of individual domains are well-separated, the domain label $d$ of a given sample $\mathbf{x}$ can be accurately predicted, and vice versa. Based on this insight, DDL encourages the domain label of a sample to be predictable from its latent feature through an auxiliary domain classifier. More specifically, we introduce a standard cross-entropy loss that requires the model to correctly predict the domain labels of the samples:

$$\mathcal{L}_{\text{CE}}(\mathcal{B}) = -\frac{1}{|\mathcal{B}|} \sum_{i=1}^{|\mathcal{B}|} \sum_{j=1}^{|\mathcal{D}|} d_{ij} \log p_{ij}^d, \tag{3}$$

where $\mathbf{p}_i^d = (p_{i1}^d, p_{i2}^d, \ldots, p_{i|\mathcal{D}|}^d)$ represents the confidence scores of a sample $\mathbf{x}_i$ output by the domain classifier (a fully connected layer), and $\mathbf{d}_i = (d_{i1}, d_{i2}, \ldots, d_{i|\mathcal{D}|})$ is the one-hot encoding of the domain label $d_i$.

To further enhance domain separability, we additionally incorporate the Maximum Mean Discrepancy (MMD) into DDL as an auxiliary loss term. MMD estimates the pairwise distance between domain distributions in a Reproducing Kernel Hilbert Space (RKHS) as:

$$\text{MMD}^2(\mathcal{B}) = \frac{2}{|\mathcal{D}|(|\mathcal{D}| - 1)} \sum_{1 \le d < d' \le |\mathcal{D}|} \left\| \frac{1}{|\mathcal{B}_d|} \sum_{\mathbf{x}_i \in \mathcal{B}_d} \phi(\mathbf{x}_i) - \frac{1}{|\mathcal{B}_d'|} \sum_{\mathbf{x}_j \in \mathcal{B}_{d'}} \phi(\mathbf{x}_j) \right\|_{\mathcal{H}}^2, \tag{4}$$

where $\phi$ denotes a kernel-induced feature mapping, $\mathcal{B}_d$ is a subset of mini-batch $\mathcal{B}$ within the domain $d \in \mathcal{D}$. Intuitively, maximizing MMD increases the inter-domain divergence in the latent space.

Given the above formulations, our final DDL loss is defined as:

$$\mathcal{L}_{\text{domain}}(\mathcal{B}) = \gamma \mathcal{L}_{\text{CE}}(\mathcal{B}) - \lambda \text{MMD}^2(\mathcal{B}), \tag{5}$$

where $\gamma$ and $\lambda$ are balancing hyperparameters. Combining with the standard loss functions $\mathcal{L}_{\text{memorize}}$ and $\mathcal{L}_{\text{forget}}$, the learnable prompts and the domain classifier are jointly optimized by minimizing the total loss:

$$\mathcal{L}_{\text{total}}(\mathcal{B}) = \mathcal{L}_{\text{memorize}}(\mathcal{B}) + \mathcal{L}_{\text{forget}}(\mathcal{B}) + \mathcal{L}_{\text{domain}}(\mathcal{B}). \tag{6}$$

### 3.4 Instance-wise Prompt Generator (InstaPG)

Domain is often ambiguous. The term "illustration," for example, encompasses a broad spectrum of styles, from highly realistic renderings that closely resemble real-world images to highly stylized depictions resembling clipart, with each image varying in style. Given this nature of the domain, only a learnable prompt that is uniform across all the images cannot account for such an instance-level variation in the images, which may degrade the performance of domain unlearning.

Table 1: **Comparison with Baseline CLIP Fine-tuning Methods.** The average performance of all possible combinations of the domains to be forgotten/retained are reported for various number of domains to be forgotten $|\mathcal{D}_{\text{forget}}| \in \{1, 2, 3\}$. ImageNet has only two domains, so only the case $|\mathcal{D}_{\text{forget}}| = 1$ is tested. Performance is evaluated using the three metrics: (i) *Mem*: the accuracy of the classes for data from memorized domains, (ii) *For*: the error of the classes for data from forgotten domains, and (iii) $H$: the harmonic mean of *Mem* and *For*. Higher values mean better performance. The *Baseline* in the table refers to the method that trains the vision prompt using $\mathcal{L}_{\text{memorize}}$ and $\mathcal{L}_{\text{forget}}$ in Sec. 3.2, representing a straightforward and conventional approach to unlearning.

| $|\mathcal{D}_{\text{forget}}|$ | Method | ImageNet | | | Office-Home | | | Mini DomainNet | | |
|---|---|---|---|---|---|---|---|---|---|---|
| | | $H \uparrow$ | $Mem\uparrow$ | $For\uparrow$ | $H \uparrow$ | $Mem\uparrow$ | $For\uparrow$ | $H \uparrow$ | $Mem\uparrow$ | $For\uparrow$ |
| $|\mathcal{D}_{\text{forget}}| = 1$ | LP++ [Huang et al., 2024a] | 50.69 | 62.22 | 42.76 | 30.46 | **85.06** | 18.55 | 31.73 | 84.72 | 19.52 |
| | CLIPFit [Li et al., 2024] | 71.31 | 63.04 | 82.13 | 43.44 | 83.20 | 29.40 | 53.56 | 84.32 | 39.24 |
| | BBF [Kuwana et al., 2024] | 45.56 | 35.92 | 65.91 | 31.25 | 80.94 | 19.74 | 32.12 | 81.80 | 20.03 |
| | Baseline | 74.66 | 61.23 | 96.43 | 52.59 | 79.96 | 39.88 | 62.07 | **85.15** | 49.40 |
| | **Ours** | **77.02** | **64.13** | **96.73** | **69.96** | 77.93 | **64.34** | **75.56** | 78.32 | **73.06** |
| $|\mathcal{D}_{\text{forget}}| = 2$ | LP++ [Huang et al., 2024a] | $\times$ | $\times$ | $\times$ | 31.11 | **85.54** | 19.01 | 32.21 | **85.29** | 19.85 |
| | CLIPFit [Li et al., 2024] | $\times$ | $\times$ | $\times$ | 40.53 | 83.64 | 26.74 | 51.35 | 84.74 | 36.83 |
| | BBF [Kuwana et al., 2024] | $\times$ | $\times$ | $\times$ | 32.54 | 80.42 | 20.54 | 47.95 | 61.48 | 40.52 |
| | Baseline | $\times$ | $\times$ | $\times$ | 54.19 | 80.23 | 41.23 | 61.97 | 84.71 | 48.96 |
| | **Ours** | $\times$ | $\times$ | $\times$ | **73.58** | 75.61 | **71.89** | **77.03** | 76.51 | **77.66** |
| $|\mathcal{D}_{\text{forget}}| = 3$ | LP++ [Huang et al., 2024a] | $\times$ | $\times$ | $\times$ | 33.57 | **86.22** | 20.84 | 34.73 | **85.87** | 21.77 |
| | CLIPFit [Li et al., 2024] | $\times$ | $\times$ | $\times$ | 50.02 | 85.68 | 35.32 | 52.88 | 85.39 | 38.30 |
| | BBF [Kuwana et al., 2024] | $\times$ | $\times$ | $\times$ | 39.60 | 78.13 | 26.80 | 39.29 | 80.29 | 26.05 |
| | Baseline | $\times$ | $\times$ | $\times$ | 59.47 | 80.96 | 48.79 | 68.82 | 84.25 | 58.27 |
| | **Ours** | $\times$ | $\times$ | $\times$ | **75.89** | 72.15 | **80.77** | **79.00** | 75.00 | **83.94** |

To address this problem, we introduce an Instance-wise Prompt Generator (InstaPG) to adjust the learnable vision prompts according to the input image patches. The illustration is given in Fig. 2b. InstaPG is embedded in an intermediate layer (i.e., Transformer block) of the image encoder to generate additional instance-wise prompts to be fed to the subsequent layer via a cross-attention mechanism, where the learnable vision prompts serve as queries, while the image patch features act as keys and values. The generated prompts are conditioned on the input image patch features, allowing the model effectively captures the property of the input image.

## 4 Experiments

### 4.1 Settings

**Datasets.** We evaluate our method on four public multi-domain image classification datasets: ImageNet [Deng et al., 2009], Office-Home [Venkateswara et al., 2017], Mini DomainNet [Zhou et al., 2021], and DomainNet [Peng et al., 2019], which are widely used for evaluating domain adaptation/generalization methods. For ImageNet, we treat ImageNet-1K [Deng et al., 2009] and ImageNet-Sketch [Wang et al., 2019] as two distinct domains, containing approximately 1.28M and 50K samples, respectively, across 1,000 object classes. Office-Home [Venkateswara et al., 2017] contains 15.5K samples of 65 object classes over four domains, namely, *art*, *clipart*, *product*, and *real-world*. Mini DomainNet [Zhou et al., 2021] consists of 140K images of 126 object classes from four domains: *clipart*, *painting*, *real*, and *sketch*. Details and results on DomainNet [Peng et al., 2019] are provided in Appendix A. Unless otherwise noted, we use eight labeled samples per domain (both class and domain labels) for training, following the few-shot setting commonly adopted in recent VLM tuning [Zhou et al., 2022b, Khattak et al., 2023, Li et al., 2024, Huang et al., 2024a] and machine unlearning studies [Kuwana et al., 2024].

**Baselines.** Since this is the first work that introduces the task of ADU, there is no existing method directly applicable to the task. Thus, we designed dedicated baselines for comparative experiments. Specifically, we evaluate the two state-of-the-art CLIP fine-tuning methods, namely LP++ [Huang et al., 2024a] and CLIPFit [Li et al., 2024], tuned with the same loss functions given in Sec. 3.2, i.e., $\mathcal{L}_{\text{memorize}}$ and $\mathcal{L}_{\text{forget}}$. We also compare our method with the state-of-the-art machine unlearning

Table 2: **Ablation Study.** The ablation results of Domain Disentangling Loss (DDL) and Instance-wise Prompt Generator (InstaPG) are reported. While our method achieves improvements with either DDL or InstaPG alone, combining both leads to even better balanced trade-off performance.

| Dataset | DDL | InstaPG | $|\mathcal{D}_{\text{forget}}| = 1$ | | | $|\mathcal{D}_{\text{forget}}| = 2$ | | | $|\mathcal{D}_{\text{forget}}| = 3$ | | |
|---|---|---|---|---|---|---|---|---|---|---|---|
| | | | $H\uparrow$ | $Mem\uparrow$ | $For\uparrow$ | $H\uparrow$ | $Mem\uparrow$ | $For\uparrow$ | $H\uparrow$ | $Mem\uparrow$ | $For\uparrow$ |
| Office-Home | - | - | 52.59 | 79.96 | 39.88 | 54.19 | 80.23 | 41.23 | 59.47 | 80.96 | 48.79 |
| | - | ✓ | 56.41 | **83.55** | 44.05 | 61.01 | **83.33** | 48.60 | 70.60 | **81.55** | 63.28 |
| | ✓ | - | 60.82 | 74.51 | 51.72 | 60.01 | 76.22 | 49.94 | 63.12 | 77.74 | 54.57 |
| | ✓ | ✓ | **69.96** | 77.93 | **64.34** | **73.58** | 75.61 | **71.89** | **75.89** | 72.15 | **80.77** |
| Mini DomainNet | - | - | 62.07 | 85.15 | 49.40 | 61.97 | 84.71 | 48.96 | 68.82 | **84.25** | 58.27 |
| | - | ✓ | 64.06 | **85.46** | 51.64 | 65.92 | **85.12** | 53.96 | 74.17 | 83.68 | 66.68 |
| | ✓ | - | 74.23 | 78.00 | 70.87 | 75.28 | 77.57 | 73.20 | 77.60 | 77.42 | 77.90 |
| | ✓ | ✓ | **75.56** | 78.32 | **73.06** | **77.03** | 76.51 | **77.66** | **81.78** | 77.96 | **86.12** |

Table 3: **Ablation Studies of Loss Functions in Domain Disentangling Loss (DDL).** DDL employs two loss functions: Cross-Entropy (CE) and Maximum Mean Discrepancy (MMD). Using both losses together yields the best-balanced trade-off, as shown by $H$.

| CE | MMD | $|\mathcal{D}_{\text{forget}}| = 1$ | | | $|\mathcal{D}_{\text{forget}}| = 2$ | | | $|\mathcal{D}_{\text{forget}}| = 3$ | | |
|---|---|---|---|---|---|---|---|---|---|---|
| | | $H\uparrow$ | $Mem\uparrow$ | $For\uparrow$ | $H\uparrow$ | $Mem\uparrow$ | $For\uparrow$ | $H\uparrow$ | $Mem\uparrow$ | $For\uparrow$ |
| - | - | 56.41 | **83.55** | 44.05 | 61.01 | **83.33** | 48.60 | 70.60 | 81.55 | 63.28 |
| - | ✓ | 68.62 | 82.41 | 59.47 | 72.21 | 81.01 | 65.66 | 66.76 | **82.38** | 57.12 |
| ✓ | - | 64.01 | 82.97 | 53.62 | 69.33 | 80.73 | 61.24 | 71.53 | 81.43 | 64.55 |
| ✓ | ✓ | **69.96** | 77.93 | **64.34** | **73.58** | 75.61 | **71.89** | **75.89** | 72.15 | **80.77** |

method for VLMs, Black-Box Forgetting (BBF) [Kuwana et al., 2024]. In addition, we evaluate a "Baseline" method that learns the learnable vision prompts with $\mathcal{L}_{\text{memorize}}$ and $\mathcal{L}_{\text{forget}}$.

**Implementation Details.** We use a pre-trained CLIP model with ViT-B/16 [Dosovitskiy et al., 2021] as the image encoder. The text prompt is set to "a photo of a [class]". For vision prompts, we adopt deep prompting [Khattak et al., 2023] with eight learnable context tokens and train the model for 50 epochs using SGD with a learning rate of 0.0025. The vision prompts are optimized within the first nine transformer layers of the image encoder. We consistently set the weights in loss functions $\gamma = 30$ and $\lambda = 10$.

**Evaluation Metrics.** We use the following three metrics: (i) *Mem*: the accuracy of the classes for data from memorized domains $\{(\mathbf{x}, y, d)|d \in \mathcal{D}_{\text{memorize}}\}$; (ii) *For*: the error of the classes for data from forgotten domains $\{(\mathbf{x}, y, d)|d \in \mathcal{D}_{\text{forget}}\}$; (iii) *H*: the harmonic mean of *Mem* and *For*, representing the overall unlearning performance as it balances the forgetting rate for domains to be forgotten and the classification accuracy for domains to be memorized. Higher values for all these metrics are desirable. We report the average results over three runs using different random seeds.

## 4.2 Results

The comparative results are shown in Table 1. Our method significantly outperforms all the compared methods on all the datasets in both $H$ and *For*.

Comparing our method with the two state-of-the-art CLIP fine-tuning methods (i.e., LP++ and CLIP-Fit), our method ourperforms them by more than $20\%$ in $H$ on Office-Home and Mini DomainNet, and by $5.71\%$ on ImageNet, regardless of the number of domains to be forgotten. Furthermore, the forgetting performance (*For*) of CLIPFit and LP++ is below $40\%$ on Office-Home and Mini DomainNet, whereas our method achieves over $60\%$. These results demonstrate that even when state-of-the-art VLM fine-tuning methods are equipped with common approximate unlearning strategies, they still struggle with ADU – highlighting the unique difficulty of domain unlearning.

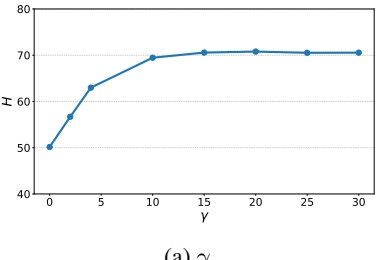

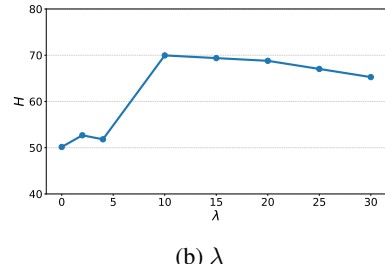

(a) $\gamma$          (b) $\lambda$

Figure 3: **Impact of Loss Weights $\gamma$ and $\lambda$ in Domain Disentangling Loss (DDL)**. We analyze the effect of varying the loss weights $\gamma$ and $\lambda$, which control the weights of the cross-entropy loss and the Maximum Mean Discrepancy (MMD) loss, respectively. Performance remains stable across a wide range of values once both $\gamma$ and $\lambda$ exceed a certain threshold, indicating that the proposed method is not highly sensitive to the choice of these hyperparameters.

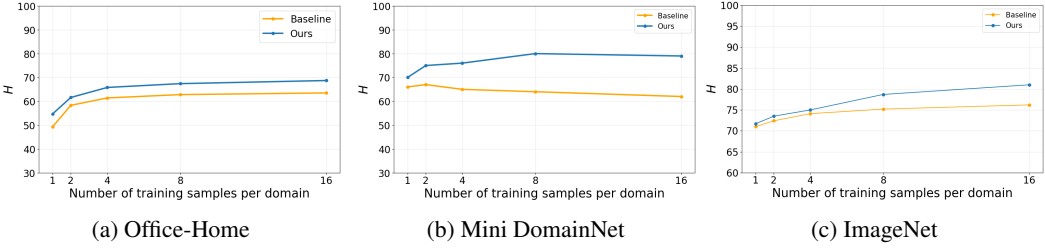

(a) Office-Home       (b) Mini DomainNet       (c) ImageNet

Figure 4: **Sensitivity to The Number of Training Samples per Domain.** We compare our method with Baseline, which uses $\mathcal{L}_{\mathrm{memorize}}$ and $\mathcal{L}_{\mathrm{forget}}$ for vision prompt learning. While Baseline shows limited improvement with more shots, especially on Mini DomainNet, our method consistently improves, demonstrating better generalization and reduced overfitting.

When compared with BBF, the current state-of-the-art class unlearning method for vision-language models applied to ADU, our method achieves over $30\%$ higher *For* on all datasets. These results suggest that class unlearning methods are not sufficient for domain unlearning, even a state-of-the-art method, and further validate the effectiveness of our approach.

The Baseline in the table refers to the method that trains the vision prompt using only the common class unlearning losses, $\mathcal{L}_{\mathrm{memorize}}$ and $\mathcal{L}_{\mathrm{forget}}$ (see Sec. 3.2), representing a straightforward and conventional approach to unlearning. On OfficeHome and Mini DomainNet, our method surpasses Baseline by over $20\%$ in *For*, demonstrating the forgetting capability of our method and showing the limitations of naive unlearning strategies in ADU.

## 4.3 Analysis

**Ablation Study.** Table 2 shows an ablation study of the proposed Domain Disentangling Loss (DDL) and Instance-wise Prompt Generator (InstaPG) on the Office-Home and Mini DomainNet datasets. Although DDL or InstaPG alone improves trade-off performance, combining it with InstaPG further boosts the scores in both $H$ and *For*. To encourage disentanglement in the latent feature space, our DDL employs Cross-Entropy (CE) and Maximum Mean Discrepancy (MMD) losses (see Eq. (5)). Table 3 shows an ablation of these two components on Office-Home. Both CE and MMD promote effective forgetting while preserving accuracy on memorized domains. Using both losses together yields the best-balanced trade-off performance, demonstrating the complementary effectiveness of CE and MMD losses within DDL.

**Sensitivity to Loss Weights.** We analyze the sensitivity of our method to two hyperparameters, $\gamma$ and $\lambda$, which respectively balance CE and MMD losses in DDL. Fig. 3 shows the effect of varying $\gamma$ and $\lambda$ on performance. As $\gamma$ increases, performance improves and saturates around $\gamma = 10$. Notably, it remains stable across a wide range of values once $\gamma$ exceeds a certain threshold, suggesting that the method is not overly sensitive to the choice of $\gamma$. A similar trend is observed for $\lambda$. Performance peaks around $\lambda = 10$, and remains stable across a wide range beyond that. This suggests that as long

Table 4: **Domain Classification Accuracy when *Art* is Forgotten on Office-Home.**

| Method | Domain Classification Accuracy [%] |
|---|---|
| Before Unlearning | 25.80 |
| Ours w/o DDL and InstaPG | 31.06 |
| Ours | 79.43 |

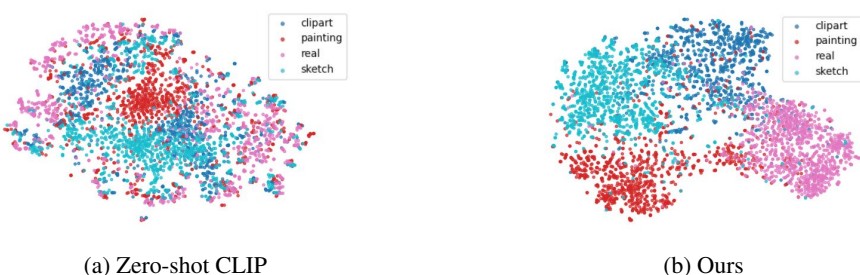

(a) Zero-shot CLIP              (b) Ours

Figure 5: **t-SNE Visualization, Where the Domain to Be Forgotten is *Real*.** (a) In the feature space of zero-shot CLIP, features from different domains are entangled, indicating poor domain separation, which makes domain-wise control over feature representations difficult. (b) By applying our method, the features are effectively disentangled from their domains, facilitating domain-wise control.

as $\lambda$ is set above a certain threshold, the exact value has limited impact, making the method relatively insensitive to the choice of $\lambda$.

**Sensitivity to Number of Training Samples.** In practical scenarios, the number of available samples per domain can vary substantially, making it important to assess how unlearning methods respond to different data regimes. We therefore evaluate the sensitivity of our approach to the number of training samples (i.e., shots) per domain, and compare it with the Baseline method. As shown in Fig. 4, our method consistently improves as the number of shots increases, demonstrating its ability to effectively leverage additional data. In contrast, Baseline struggles to benefit from more shots on Mini DomainNet, suggesting a tendency toward overfitting. Our method continues to gain performance with additional training samples, highlighting its robustness and stronger generalization capability.

**Domain Classification Accuracy.** An essential prerequisite for effective domain unlearning is that feature representations from different domains are well separated; in other words, domains should be easily distinguishable. Otherwise, attempts to suppress recognition accuracy on a specific domain would inevitably affect others. Therefore, evaluating not only the final unlearning performance but also the domain classification accuracy is crucial, as it provides direct evidence of the effectiveness of our key components, namely, DDL and InstaPG, in enhancing domain separability. We report domain classification accuracy before and after unlearning on Office-Home in Table 4. The accuracy increases substantially from 25.80% before unlearning to 79.43% after applying our method. When the DDL and InstaPG are removed, the accuracy drops remarkably to 31.06%, highlighting the critical role of these components in achieving effective domain separation. These findings provide strong additional evidence of the effectiveness of our method, particularly in facilitating domain-level unlearning.

**t-SNE Visualization.** Fig. 5 presents a t-SNE visualization of image features from Mini DomainNet, where the domain to forget is *real*. In Fig. 5a, the features extracted by Zero-shot CLIP are heavily entangled across domains, indicating poor domain separation, which makes domain-wise control difficult. Fig. 5b demonstrates that our method separates the domains in the feature space, enabling control over memorization and forgetting on per-domain basis.

**Attention Map Visualization.** To further investigate and understand the behavior of our unlearned model, we show the attention heatmaps [Selvaraju et al., 2017] before and after applying our method on DomainNet Mini dataset in Fig. 6, where the domain to be forgotten is *real*. For the forgetting data, the Zero-shot CLIP attention concentrates on the objects. After applying our method, the attention on the objects disappears or significantly weakens for the data from the domain to be forgotten (i.e., *real*). For the data from the domain to be memorized (i.e., *painting*, *clipart*, and *sketch*), our method fully maintains or strengthens previous attention on the objects. Our method suppresses

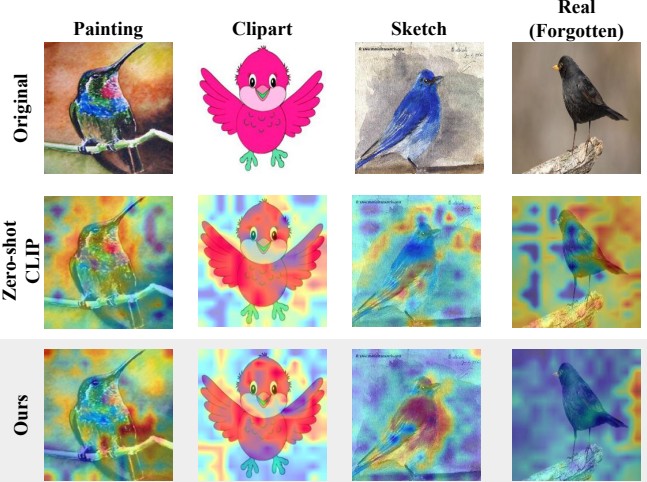

Figure 6: **Visualization of Attention Maps.** From top to bottom: original image, Zero-shot CLIP attention, and unlearned model attention by our method, where the domain to be forgotten is *real*. Zero-shot CLIP attention concentrates on objects regardless of domains, demonstrating its generalizability across various domains. Our method successfully distracts the model's attention from the semantic regions of the forgetting data, while preserving it for the remaining data.

prediction sensitivity for data from domains to be forgotten while preserving or enhancing sensitivity for data from domains to be memorized, enabling the model to effectively forget unwanted domain information while maintaining high accuracy on the retained domains.

## 5   Limitations

Our method performs Approximate Domain Unlearning (ADU) under the assumption that domain labels are available for all training samples. While this setting enables explicit and fine-grained control over domain-specific forgetting, it may not always hold in real-world scenarios, where such comprehensive domain information is often incomplete or unavailable. Nevertheless, this limitation may be mitigated by integrating domain estimation techniques, which can estimate domain clusters without requiring prior knowledge of the domains. Indeed, we present additional experiments in Appendix C.3, where we simulate missing domain labels and evaluate the effectiveness of incorporating domain estimation through a simple pseudo-labeling technique. The results show that, even when a large fraction of domain labels is absent, our method maintains substantially better performance with domain estimation than without it. This suggests that combining ADU with more advanced domain estimation techniques (e.g., [Mitsuzumi et al., 2021]) provides a practical path toward handling more realistic settings with incomplete domain information. Although such integration lies beyond the scope of this work, exploring this direction offers a promising avenue for realizing ADU under more challenging conditions.

## 6   Conclusions

We introduced *Approximate Domain Unlearning (ADU)*, a brand new variant of approximate unlearning that prevents pre-trained Vision-Language Models (VLMs) from recognizing only data in specified domains. We also devised a novel solution to this task based on the idea of allowing domain unlearning by disentangling the strong domain generalizability of pre-trained VLMs. Experimental results showed that our method outperformed the dedicated baselines. We believe this paper opens up a new research direction on approximate unlearning and provides a new challenge to the community.

## 7   Acknowledgements

This paper is partially based on results obtained from a project, JPNP25006, commissioned by the New Energy and Industrial Technology Development Organization (NEDO).

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

# A  Results on DomainNet

We evaluated our method on DomainNet [Peng et al., 2019], an established multi-domain benchmark dataset that comprises 177K test samples across six domains and 345 classes, making it one of the most complex and large-scale datasets of its kind. The results are shown in Table 5. Ours is clearly better than Baseline, which emphasizes the strong effectiveness of our method.

Table 5: **Results on DomainNet.**

| Method | $|\mathcal{D}_{\text{forget}}| = 1$ | | | $|\mathcal{D}_{\text{forget}}| = 2$ | | | $|\mathcal{D}_{\text{forget}}| = 3$ | | |
|---|---|---|---|---|---|---|---|---|---|
| | $H \uparrow$ | $Mem\uparrow$ | $For\uparrow$ | $H \uparrow$ | $Mem\uparrow$ | $For\uparrow$ | $H \uparrow$ | $Mem\uparrow$ | $For\uparrow$ |
| Zero-shot CLIP | 36.85 | 53.28 | 28.16 | 36.85 | 53.28 | 28.16 | 36.85 | 53.28 | 28.16 |
| Baseline | 38.45 | 55.69 | 29.36 | 39.36 | 55.81 | 30.40 | 38.79 | 55.26 | 29.88 |
| Ours | **66.81** | **58.86** | **77.23** | **67.81** | **58.89** | **79.90** | **68.83** | **59.06** | **82.48** |

# B  Per-Domain Accuracy

We report per-domain accuracy on Office-Home, Mini DomainNet, and DomainNet, which include domains with substantial visual similarity (e.g., *sketch* vs. *quickdraw*). The results are shown in Tables 6, 7, and 8. Overall, they demonstrate that our method can selectively suppress classification accuracy on the forgotten domain while maintaining performance on the others, even when domains share similar visual characteristics. For instance, in Mini DomainNet, forgetting *sketch* reduces its accuracy significantly from 72.54% to 20.64%, whereas visually similar domains such as *clipart* and *painting* are only minimally affected (within 1%). These results indicate that our method offers fine-grained control even under significant domain overlap.

We found that unlearning can be more difficult for certain domains. For example, when forgetting *real* in Office-Home, our method decreases the accuracy from 81.29% to 55.48%. Although this is substantially lower than the original zero-shot CLIP, the model still retains moderate recognition ability. One possible reason is that CLIP is heavily pre-trained on image-text pairs dominated by real-world photos, making the *real* domain more strongly encoded and thus harder to forget. Nonetheless, as shown in Table 1 of the main paper, our method consistently achieves strong forgetting performance on average across datasets. These results suggest that while our approach is broadly effective, slight variations may arise depending on the VLM's pretraining bias; mitigating dataset bias could serve as a potential remedy.

Table 6: **Per-Domain Accuracy on Office-Home.**

| Method | *art* | *clipart* | *product* | *real* |
|---|---|---|---|---|
| Zero-shot CLIP | 74.34 | 60.97 | 80.43 | 81.29 |
| Ours ($\mathcal{D}_{\text{forget}} = \{art\}$) | **39.25** | 70.18 | 88.72 | 75.91 |
| Ours ($\mathcal{D}_{\text{forget}} = \{clipart\}$) | 77.36 | **15.13** | 87.87 | 80.00 |
| Ours ($\mathcal{D}_{\text{forget}} = \{product\}$) | 80.38 | 72.15 | **32.77** | 77.85 |
| Ours ($\mathcal{D}_{\text{forget}} = \{real\}$) | 68.30 | 67.98 | 78.94 | **55.48** |

Table 7: **Per-Domain Accuracy on Mini DomainNet.**

| Method | *clipart* | *painting* | *real* | *sketch* |
|---|---|---|---|---|
| Zero-shot CLIP | 80.64 | 78.10 | 87.94 | 72.54 |
| Ours ($\mathcal{D}_{\text{forget}} = \{clipart\}$) | **24.76** | 75.40 | 83.97 | 73.97 |
| Ours ($\mathcal{D}_{\text{forget}} = \{painting\}$) | 81.59 | **30.32** | 84.92 | 73.97 |
| Ours ($\mathcal{D}_{\text{forget}} = \{real\}$) | 77.94 | 69.52 | **32.06** | 74.44 |
| Ours ($\mathcal{D}_{\text{forget}} = \{sketch\}$) | 79.05 | 77.62 | 87.46 | **20.64** |

Table 8: **Per-Domain Accuracy on DomainNet.**

| Method | clipart | infograph | painting | quickdraw | real | sketch |
|---|---|---|---|---|---|---|
| Zero-shot CLIP | 71.84 | 50.01 | 65.36 | 14.91 | 83.42 | 63.07 |
| Ours ($\mathcal{D}_{\text{forget}} = \{clipart\}$) | **33.37** | 51.55 | 67.72 | 28.93 | 79.87 | 62.22 |
| Ours ($\mathcal{D}_{\text{forget}} = \{infograph\}$) | 71.84 | **13.05** | 67.77 | 32.07 | 79.50 | 63.55 |
| Ours ($\mathcal{D}_{\text{forget}} = \{painting\}$) | 73.59 | 52.21 | **22.77** | 29.64 | 79.86 | 63.21 |
| Ours ($\mathcal{D}_{\text{forget}} = \{quickdraw\}$) | 73.60 | 51.33 | 67.99 | **7.01** | 81.79 | 63.79 |
| Ours ($\mathcal{D}_{\text{forget}} = \{real\}$) | 71.61 | 50.52 | 64.40 | 29.56 | **35.89** | 64.05 |
| Ours ($\mathcal{D}_{\text{forget}} = \{sketch\}$) | 70.81 | 51.62 | 67.20 | 29.62 | 81.26 | **24.51** |

Table 9: **Robustness to Domain Imbalance on Office-Home.**

| Number of shots for selected domains | $|\mathcal{D}_{\text{forget}}| = 1$ | | | $|\mathcal{D}_{\text{forget}}| = 2$ | | | $|\mathcal{D}_{\text{forget}}| = 3$ | | |
|---|---|---|---|---|---|---|---|---|---|
| | H ↑ | Mem↑ | For↑ | H ↑ | Mem↑ | For↑ | H ↑ | Mem↑ | For↑ |
| 8 shots | 69.96 | 77.93 | 64.34 | 73.58 | 75.61 | 71.89 | 75.89 | 72.15 | 80.77 |
| 4 shots | 68.23 | 82.24 | 59.55 | 71.24 | 81.21 | 63.82 | 74.34 | 74.61 | 74.08 |
| 1 shot | 66.89 | 77.64 | 59.61 | 63.80 | 78.16 | 56.53 | 68.75 | 64.31 | 73.85 |

## C   Robustness to More Complex Scenarios

In addition to the standard settings, we further evaluate the robustness of our method under more complex and realistic scenarios that may arise in practice. Specifically, we consider three challenging conditions: (i) domain imbalance, where some domains contain far fewer samples than others; (ii) partial domain-class overlap, where certain classes appear only in a subset of domains; and (iii) partial domain labels, where domain annotations are missing for a portion of the training samples. These experiments aim to validate whether our approach remains effective in handling such imbalanced, heterogeneous, and incomplete settings.

### C.1   Robustness to Domain Imbalance

Domain imbalance can arise in practical scenarios, where certain domains have substantially fewer samples than others (e.g., abundant *real* images vs. sparse *art* or *clipart* images in autonomous driving). To assess the robustness of our DDL under such imbalance, we conducted experiments on Office-Home by reducing the number of training samples from selected domains (*art* and *clipart*), while keeping the other domains fixed. Our default setting uses eight shots, and we additionally tested reduced settings with four or one shot(s).

As shown in Table 9, our method maintains stable performance even with severe imbalance. Remarkably, with only a single sample from the *art* and *clipart* domains, the model still achieves competitive performance. These results indicate that our method can still perform effective domain disentangling and selective forgetting even in highly imbalanced scenarios.

### C.2   Robustness to Partial Domain-Class Overlap

We also investigated a scenario where domain-class distributions are partially overlapping. Specifically, in Office-Home we removed samples from three random classes only in the *art* and *clipart* domains, thereby creating a setting where some classes are missing in certain domains. As presented in Table 10, our method retains competitive performance under this condition. These results suggest that our method can robustly handle realistic domain-class imbalances without a critical loss in effectiveness.

### C.3   Robustness to Partial Domain Labels

As we discuss in Sec. 5, we acknowledge that domain labels may not always be available for all training samples in real-world scenarios. However, it is important to note that most standard formulations in domain adaptation and domain generalization (e.g., [Wu et al., 2024, Cho et al., 2023,

Table 10: **Robustness to Partial Domain-Class Overlap on Office-Home.**

| Setting | $|\mathcal{D}_{\text{forget}}| = 1$ | | | $|\mathcal{D}_{\text{forget}}| = 2$ | | | $|\mathcal{D}_{\text{forget}}| = 3$ | | |
|---|---|---|---|---|---|---|---|---|---|
| | $H\uparrow$ | $Mem\uparrow$ | $For\uparrow$ | $H\uparrow$ | $Mem\uparrow$ | $For\uparrow$ | $H\uparrow$ | $Mem\uparrow$ | $For\uparrow$ |
| Original Dataset | **69.96** | 77.93 | **64.34** | **73.58** | 75.61 | **71.89** | **75.89** | 72.15 | **80.77** |
| Dataset w/ Partial Domain-Class Overlap | 67.93 | **81.81** | 58.69 | 72.03 | **79.61** | 66.24 | 74.51 | **75.12** | 73.90 |

Table 11: **Robustness to Partial Domain Labels on Office-Home.** Without domain estimation, the performance drops sharply as the proportion of unlabeled samples increases. In contrast, with domain estimation, the performance remains substantially higher,

| Unlabeled sample ratio | w/o Domain Estimation | | | w/ Domain Estimation | | |
|---|---|---|---|---|---|---|
| | $H\uparrow$ | $Mem\uparrow$ | $For\uparrow$ | $H\uparrow$ | $Mem\uparrow$ | $For\uparrow$ |
| 0.0 | 75.89 | 72.15 | 80.77 | 75.89 | 72.15 | 80.77 |
| 0.3 | 63.38 | 83.35 | 52.68 | **65.81** | 83.06 | 55.98 |
| 0.5 | 55.77 | 84.63 | 43.66 | **63.64** | 84.69 | 52.20 |
| 0.7 | 49.97 | 83.84 | 36.98 | **61.21** | 83.37 | 50.00 |

Jhoo and Heo, 2021]) assume full access to domain labels during training. Therefore, our setting aligns with this widely accepted convention and should not be considered an unrealistic simplification. Meanwhile, to evaluate the performance of our method under partial domain label availability, we consider a setting in which a portion of the training samples (ranging from 30% to 70%) lack domain annotations. To assess the effectiveness of combining domain estimation with our method in such cases, we adopt a simple pseudo-labeling approach: a domain classifier is trained using only the samples with known domain labels and then used to assign pseudo labels to the unlabeled samples. As shown in Table 11, without domain estimation, the performance drops sharply as the proportion of unlabeled samples increases. In contrast, with domain estimation, the performance remains substantially higher, even under 70% missing domain labels. These results support our claim that our approach can be extended to more realistic cases where only partial domain annotations are available.

## D Computational Complexity

Table 12 summarizes GPU memory usage and training time with an NVIDIA RTX A4000 GPU on Office-Home. While our method incurs only slightly higher computational cost than lightweight baselines [Zhou et al., 2022b, Huang et al., 2024a], it remains on par with advanced CLIP fine-tuning methods [Khattak et al., 2023, Li et al., 2024]. These results prove that our method is sufficiently efficient for practical use. Notably, adding InstaPG increases memory usage by only 1 GB and training time by less than 1 minute, indicating minimal overhead.

## E Additional Analysis

Beyond the main experiments, we conduct a series of additional analyses to further examine the robustness, sensitivity, generality, and real-world applicability of our method. Specifically, we investigate (i) ablation studies of the loss functions; (ii) sensitivity to the kernel choice in MMD; (iii) sensitivity to prompt depth; (iv) performance with a different pre-trained VLM; (v) comparison between prompt tuning and parameter tuning approaches; (vi) accuracy over training; and (vii) performance on vehicle-related data. Together, these analyses provide a more comprehensive understanding of the factors influencing the effectiveness of our method and its practical applicability.

### E.1 Ablation Study of Loss Functions

As described in Eq. (6) of the main paper, our total loss combines three terms: $\mathcal{L}_{\text{memorize}}$, $\mathcal{L}_{\text{forget}}$, and $\mathcal{L}_{\text{domain}}$, which are jointly optimized to balance memorization, forgetting, and domain separation. To gain deeper insights into their individual roles, we conduct ablation studies where each loss function

Table 12: **Computational Complexity on Office-Home.**

| Method | Memory [GB] | Time [s] |
|---|---|---|
| CoOp [Zhou et al., 2022b] | 1.9 | 283 |
| MaPLe [Khattak et al., 2023] | 10.4 | 539 |
| BBF [Kuwana et al., 2024] | 2.5 | 2882 |
| CLIPFit[Li et al., 2024] | 11.0 | 340 |
| LP++[Huang et al., 2024a] | 3.8 | 35 |
| Ours w/o InstaPG | 9.7 | 501 |
| Ours | 10.7 | 550 |

Table 13: **Ablation Study of Loss functions.** We evaluate the individual and combined impact of $\mathcal{L}_{\text{memorize}}$, $\mathcal{L}_{\text{forget}}$, and $\mathcal{L}_{\text{domain}}$ on memorization (*Mem*), forgetting (*For*), and the overall balance metric $H$. The results show that each loss function contributes to different aspects of performance, and their combination achieves the best trade-off.

| $\mathcal{L}_{\text{memorize}}$ | $\mathcal{L}_{\text{forget}}$ | $\mathcal{L}_{\text{domain}}$ | $|\mathcal{D}_{\text{forget}}| = 1$ | | | $|\mathcal{D}_{\text{forget}}| = 2$ | | | $|\mathcal{D}_{\text{forget}}| = 3$ | | |
|---|---|---|---|---|---|---|---|---|---|---|---|
| | | | $H \uparrow$ | $Mem \uparrow$ | $For \uparrow$ | $H \uparrow$ | $Mem \uparrow$ | $For \uparrow$ | $H \uparrow$ | $Mem \uparrow$ | $For \uparrow$ |
| ✓ | - | - | 27.52 | **86.22** | 16.92 | 30.11 | **86.60** | 18.36 | 36.15 | **86.20** | 22.88 |
| - | ✓ | - | 5.37 | 2.76 | 97.36 | 4.58 | 2.35 | 97.70 | 4.92 | 2.53 | 97.52 |
| - | - | ✓ | 60.82 | 74.51 | 51.72 | 60.01 | 76.22 | 49.94 | 63.12 | 77.74 | 54.57 |
| ✓ | ✓ | - | 52.59 | 79.96 | 39.88 | 54.19 | 80.23 | 41.23 | 59.47 | 80.96 | 48.79 |
| ✓ | - | ✓ | 48.64 | 81.41 | 38.52 | 57.18 | 79.92 | 45.98 | 65.03 | 79.17 | 55.46 |
| - | ✓ | ✓ | 2.74 | 1.39 | **98.83** | 2.77 | 1.41 | **98.54** | 2.68 | 1.36 | **98.37** |
| ✓ | ✓ | ✓ | **69.96** | 77.93 | 64.34 | **73.58** | 75.61 | 71.89 | **75.89** | 72.15 | 80.77 |

is applied separately or in combination. This analysis allows us to disentangle their contributions to different aspects of performance and to verify that the full combination provides the best trade-off.

Table 13 presents the results. The loss function $\mathcal{L}_{\text{memorize}}$ predominantly enhances memorization performance (*Mem*), while $\mathcal{L}_{\text{forget}}$ and $\mathcal{L}_{\text{domain}}$ mainly contribute to forgetting performance (*For*). Combining all three loss functions yields the best overall balance between memorization and forgetting, as reflected by the harmonic mean $H$.

## E.2 Choice of Kernel for MMD

Since our DDL is based on MMD, its effectiveness could in principle be sensitive to the choice of kernel. To assess sensitivity of our method to kernel choice, we tested four different kernels for MMD on Office-Home, as shown in Table 14. The results show that the linear, Laplacian, and RBF kernels yield generally stable performance, with only modest differences across metrics, while the polynomial kernel performs notably worse. Given these observations, we adopt the RBF kernel in the main experiments, but when validation-based kernel selection is feasible, any of the three stable kernels can be chosen; otherwise, the linear kernel provides a conservative and reliable default.

## E.3 Sensitivity to Prompt Depth

The depth at which prompts are inserted is an important design choice, as it determines how much information from intermediate representations is influenced by the prompts. To examine the sensitivity of our method to this factor, we vary the insertion layer of the InstaPG, while ensuring that the standard vision prompt is applied up to the same layer.

As shown in Fig. 7, the performance remains stable across different depths. This indicates that our method is robust to the choice of insertion point, reducing the need for extensive hyperparameter tuning and simplifying practical deployment.

Table 14: **Impact of Kernel Choice on Office-Home.**

| Kernel | $|\mathcal{D}_{\text{forget}}| = 1$ | | | $|\mathcal{D}_{\text{forget}}| = 2$ | | | $|\mathcal{D}_{\text{forget}}| = 3$ | | |
| --- | --- | --- | --- | --- | --- | --- | --- | --- | --- |
| | $H \uparrow$ | $Mem\uparrow$ | $For\uparrow$ | $H \uparrow$ | $Mem\uparrow$ | $For\uparrow$ | $H \uparrow$ | $Mem\uparrow$ | $For\uparrow$ |
| Linear | 66.83 | 79.85 | 58.43 | 69.32 | 77.35 | 63.46 | 73.91 | 74.02 | 74.50 |
| Laplacian | 68.50 | 79.00 | 61.44 | 71.18 | 75.93 | 67.17 | 73.66 | 71.69 | 76.54 |
| Polynomial | 36.53 | 23.29 | 84.68 | 49.73 | 41.79 | 61.40 | 53.65 | 44.49 | 67.56 |
| RBF | 69.96 | 77.93 | 64.34 | 73.58 | 75.61 | 71.89 | 75.89 | 72.15 | 80.77 |

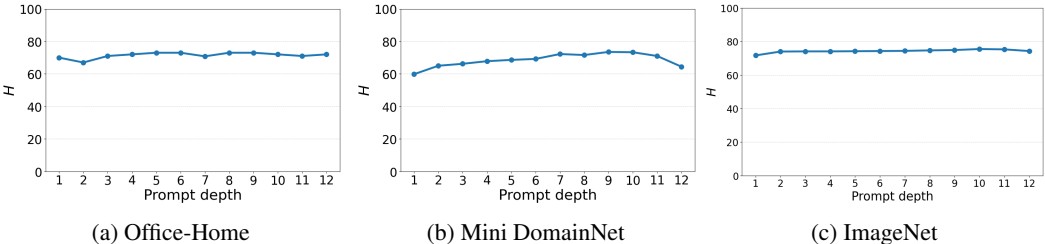

(a) Office-Home     (b) Mini DomainNet     (c) ImageNet

Figure 7: **Sensitivity to Propmt Depth.** We report the performance when varying the depth at which the InstaPG is inserted. The standard vision prompt is also applied up to the same layer. The stable results across depths demonstrate the robustness of our method to the choice of insertion point.

## E.4 Performance with Different Pre-trained VLMs

We also verify the generality of our method beyond CLIP by evaluating it on another pre-trained vision-language model, SigLIP [Zhai et al., 2023], using ImageNet. As shown in Table 15, our method outperforms Baseline, confirming that it is effective even when applied to a different backbone. This result suggests that our approach is not tied to CLIP and can be extended to other pre-trained VLMs.

## E.5 Prompt Tuning vs. Parameter Tuning

Prompt tuning with frozen model parameters has become a widely adopted alternative to full fine-tuning, as prior studies have shown that updating all parameters of CLIP often leads to catastrophic forgetting, especially in few-shot regimes [Zhou et al., 2022b, Kumar et al., 2022]. One possible concern, however, is that updating only the vision prompts might limit the ability to sufficiently alter internal representations if domain-specific features are already entangled in early layers. To examine this, we also evaluate a setting where the model parameters are updated alongside the prompts. As shown in Table 16, this setting results in severe catastrophic forgetting, whereas prompt tuning preserves a much better balance between memorization and forgetting. These findings support prompt tuning as a more reliable strategy for domain unlearning.

## E.6 Accuracy over Training Iterations

To better understand the learning dynamics of our method, we track how the memorization (*Mem*) and forgetting (*For*) accuracies evolve during training. Fig. 8 shows the results on the Office-Home dataset. We see that *Mem* drops in the early stages while *For* increases rapidly while suggesting a conflict between the two. However, both scores improve steadily after this phase, indicating that the model gradually learns to balance forgetting and retention. These results suggest a two-stage process: initial domain disentanglement followed by controlled forgetting and memorization.

Table 15: **Performance with SigLIP on ImageNet.**

| Method | $H \uparrow$ | $Mem\uparrow$ | $For\uparrow$ |
|---|---|---|---|
| Zero-shot | 46.47 | **63.26** | 36.73 |
| Baseline | 48.98 | 60.32 | 41.23 |
| Ours | **64.97** | 48.42 | **98.71** |

Table 16: **Prompt Tuning vs. Parameter Tuning on Office-Home.**

| Tuning Approach | $|\mathcal{D}_{\text{forget}}| = 1$ | | | $|\mathcal{D}_{\text{forget}}| = 2$ | | | $|\mathcal{D}_{\text{forget}}| = 3$ | | |
|---|---|---|---|---|---|---|---|---|---|
| | $H \uparrow$ | $Mem\uparrow$ | $For\uparrow$ | $H \uparrow$ | $Mem\uparrow$ | $For\uparrow$ | $H \uparrow$ | $Mem\uparrow$ | $For\uparrow$ |
| Parameter Tuning | 3.90 | 2.00 | 98.31 | 2.95 | 1.50 | 98.52 | 3.64 | 1.86 | 98.26 |
| Prompt Tuning (Ours) | **66.81** | **58.86** | **77.23** | **67.81** | **58.89** | **79.90** | **68.83** | **59.06** | **82.48** |

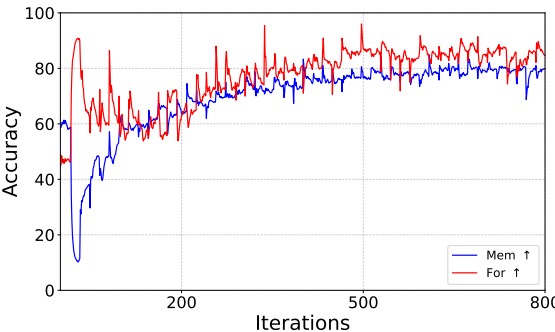

Figure 8: *Mem* and *For* over Training Iterations on Office-Home.

### E.7 Proxy Evaluation for Autonomous Driving Scenarios

To examine the practical applicability of our method in autonomous driving scenarios, we design a proxy experiment that simulates domain discrepancies commonly encountered in such systems; in real-world driving environments, vehicles may appear not only as physical objects but also as illustrations, for example in roadside advertisements and billboards (see Sec. 1). While no public autonomous driving dataset currently provides annotated examples of such illustrated vehicles, we emulate this situation using the Mini DomainNet dataset. Specifically, we select seven vehicle-related categories, namely, "bus", "car", "motorbike", "pickup truck", "police car", "school bus", and "tractor", and evaluate our method in a setting where the model is unlearned to retain only the *real* domain while forgetting the other illustration domains, i.e., *clipart*, *painting*, and *sketch*.

The results are shown in Table 17. Our method effectively preserves recognition of real vehicles, while substantially suppressing classification on illustrated ones. These results suggest that our approach holds promise for practical applications such as autonomous driving or car counting.

We acknowledge that full-scale validation on real-world deployment tasks remains an important direction for future work. Nonetheless, we believe our findings serve as an informative first step toward domain-specific unlearning under realistic constraints.

## F   Instance-level Diversity within Domain

We present example images from the *art* domain of Office-Home. Even within the same domain, the visual styles vary significantly from image to image. For instance, (a) appears to be a mix of photo and artwork. Images (b), (c), and (d) resemble typical art-style images, while (e) and (f) look more like clipart or cartoons. In contrast, (g) and (h) resemble sketch. This diversity in style within a single domain motivates the use of the Instance-wise Prompt Generator (InstaPG), which generates tailored prompts for each individual image.

Table 17: **Per-Domain Accuracy on Vehicle-Related Classes in Mini DomainNet.**

| Per-domain accuracy | clipart ↓ | painting ↓ | real ↑ | sketch ↓ |
|---|---|---|---|---|
| Before unlearning | 80.64 | 78.10 | **87.94** | 72.54 |
| Ours w/o DDL and InstaPG | 31.27 | 34.29 | 82.54 | 16.03 |
| Ours | **22.38** | **21.43** | 84.44 | **14.29** |

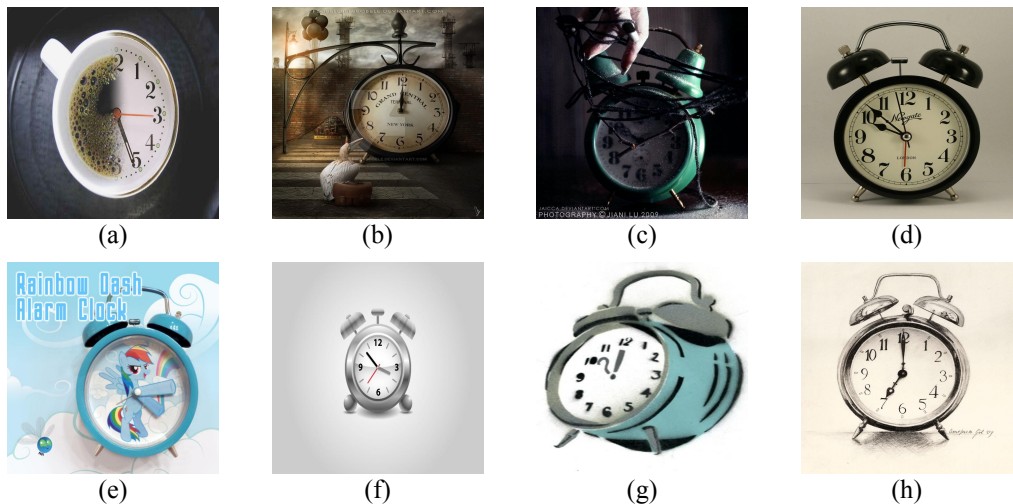

(a)     (b)     (c)     (d)

(e)     (f)     (g)     (h)

Figure 9: **Instance-level Diversity within *Art* Domain of Office-Home.** Sample images from the *art* domain exhibit substantial style variation, ranging from artwork to sketch. This intra-domain diversity motivates the use of the InstaPG, which generates image-specific prompts to better handle such variability.

