# OpenReview forum: "Approximate Domain Unlearning for Vision-Language Models"
_NeurIPS.cc/2025/Conference — NeurIPS 2025 spotlight_

### Official Review · Reviewer_ubkk · 2025-07-02

**Clarity:** 2
**Significance:** 2
**Originality:** 3
**Rating:** 4
**Confidence:** 4

**Summary:**

This paper introduces Approximate Domain Unlearning (ADU), a novel task aiming to enable pre-trained Vision-Language Models (VLMs) to selectively forget recognition capabilities for specified domains (e.g., illustrations, paintings) while preserving accuracy for target domains (e.g., real images). Conventional class unlearning methods fail to address feature entanglement caused by strong domain generalization in VLMs. The authors propose Domain Disentangling Loss (DDL) and an Instance-wise Prompt Generator (InstaPG), which enforce domain feature separation and adaptively capture instance-level variations, respectively. Experiments on three multi-domain datasets validate the method’s effectiveness

**Questions:**

1.Can the authors supplement experiments with partial domain labels to validate adaptability in real scenarios?
2.Test the method on autonomous driving datasets to verify forgetting effects on complex scenes like roadside advertisements.
3.Provide comparisons of training time and GPU resource consumption with baselines to clarify engineering costs.
4.Using DDL will reduce accuracy on memorized domains,why and how to solve it?

**Ethical Concerns:**

["NO or VERY MINOR ethics concerns only"]

**Limitations:**

yes

**Quality:**

2

**Strengths And Weaknesses:**

Strengths:
1.First to define domain-level unlearning (ADU), filling the application gap of traditional class unlearning, particularly in scenarios like autonomous driving
2.DDL maximizes inter-domain feature distances via cross-entropy and MMD, while InstaPG generates instance-specific prompts to handle intra-domain style variations, supported by solid theoretical foundations.
Weaknesses:
Dependence on Domain Labels: The method assumes complete domain labels during training, without validating robustness under partial label scenarios.
Limited Real-world Validation: Untested on practical datasets (e.g., autonomous driving), leaving domain generalization in real applications unproven.
Unclear Computational Efficiency: No comparison of training time or parameter scale with baselines, hindering evaluation of engineering feasibility.

---

> ### Author Rebuttal · Authors · 2025-07-29
>
> We sincerely thank Reviewer ubkk for their thoughtful and constructive review. We are especially grateful that the reviewer **acknowledged all the main contributions of our paper**, including the novelty of the problem setting and the effectiveness of our proposed solution. We believe this paper represents the first attempt at Approximate Domain Unlearning (ADU) in Vision-Language Models (VLMs) and offers a meaningful contribution by tackling key challenges in this emerging area. Below, we provide detailed, point-by-point responses to all questions and concerns raised. We appreciate the constructive feedback, which we believe improve our work.
>
> ### **Q1. Dependence on Domain Labels.**
>
> As we discuss in the Limitations section of our paper, we acknowledge that domain labels may not always be available for all training samples in real-world scenarios. However, it is important to note that most standard formulations in domain adaptation and domain generalization (e.g., [a–c]) assume full access to domain labels during training. Therefore, our setting aligns with this widely accepted convention and should not be considered an unrealistic simplification.
>
> Meanwhile, to evaluate the performance of our method under partial domain label availability, we consider a setting in which a portion of the training samples (ranging from 30% to 70%) lack domain annotations.
> To assess the effectiveness of combining domain estimation with our method in such cases, we adopt a simple pseudo-labeling approach: a domain classifier is trained using only the samples with known domain labels and then used to assign pseudo labels to the unlabeled samples.
>
> As shown in Table D-1, without domain estimation, the $H$-score drops from 75.89% to 49.97% under 70% missing domain labels. In contrast, with domain estimation, the $H$-score remains substantially higher at 61.21% even under the same missing rate. These results support our claim that our approach can be extended to more realistic cases where only partial domain annotations are available.
>
> *Table D-1. Performance on Office-Home with Partial Domain Labels.*
>
> | | |w/o Domain Estimation| | | w/ Domain Estimation| |
> |:---:|:---:|:---:|:---:|:---:|:---:|:---:|
> |Unlabeled sample ratio|$H\uparrow$|$Mem\uparrow$|$For\uparrow$|$H\uparrow$|$Mem\uparrow$|$For\uparrow$|
> |0|75.89|72.15|80.77|75.89|72.15|80.77|
> |0.3|**63.38**|83.35|52.68|**65.81**|83.06|55.98|
> |0.5|**55.77**|84.63|43.66|**63.64**|84.69|52.20|
> |0.7|**49.97**|83.84|36.98|**61.21**|83.37|50.00|
>
> ### **Q2: Limited Real-world Validation.**
>
> To the best of our knowledge, no public autonomous driving dataset contains annotated roadside advertisements and multi-domain versions of objects like vehicles, making direct empirical evaluation infeasible at this time.
>
> Instead, the best possible evaluation we can currently perform is a proxy experiment that partially simulates such scenarios using an existing multi-domain dataset, Mini DomainNet. Specifically, we focus on seven vehicle-related classes (bus, car, motorbike, pickup truck, police car, school bus, and tractor), and evaluate our method when retaining only the "real" domain while forgetting the other illustration domains ("clipart," "painting," and "sketch"). As shown in Table D-2, our method effectively preserves recognition of real vehicles while substantially suppressing classification on illustrated ones. These results suggest that our approach holds promise for practical applications such as autonomous driving or car counting.
>
> We acknowledge that full-scale validation on real-world deployment tasks remains an important direction for future work. Nonetheless, we believe our findings serve as an informative first step toward domain-specific unlearning under realistic constraints.
>
> *Table D-2. Per-domain Accuracy on Vehicle-Related Classes in Mini DomainNet.*
>
> |Per-domain accuracy|clipart ↓|painting ↓|real ↑|sketch ↓|
> |:----|:----|:----|:----|:----|
> |Before unlearning|80.64|78.10|**87.94**|72.54|
> |Ours w/o DDL and InstaPG|31.27|34.29|82.54|16.03|
> |Ours|**22.38**|**21.43**|84.44|**14.29**|
>
> ### **Q3. Unclear Computational Efficiency.**
>
> Table D-3 below summarizes GPU memory usage and training time with NVIDIA RTX A4000 GPU on Office-Home. While our method incurs only slightly higher computational cost than lightweight baselines such as CoOp and LP++, it remains on par with advanced CLIP fine-tuning methods such as MaPLe and CLIPFit. These results prove that our method is sufficiently efficient for practical use. Notably, adding InstaPG increases memory usage by only 1 GB and training time by just 1 minute, indicating minimal overhead.
>
> *Table D-3. Computational Complexity.*
>
> | |Memory [GB]|Time [s]|
> |:----|:----|:----|
> |CoOp [a]|1.9 | 283 |
> |MaPLe [b]| 10.4 | 539 |
> |BBF|2.5| 2882 |
> |CLIPFit| 11.0 | 340 |
> |LP++| 3.8 | 35 |
> |Ours w/o  InstaPG |9.7| 501 |
> |Ours |10.7| 550 |
>
> ### **Q4. DDL reduce accuracy on memorized domains?**
>
> Partially true. As shown in Table D-4, the per-domain accuracy on Office-Home illustrates that DDL can have varying effects across domains. For example, when "art" is forgotten, the accuracy for the retained domain "real" slightly drops from 81.29% to 75.91%, whereas the accuracy for "product" improves from 80.43% to 88.72%. These results indicate that the effect of DDL depends on inter-domain correlations and task structure, leading to either small gains or small losses in retained domains. Note that such degradations are modest and do not lead to critical failure. Overall, DDL serves as an effective mechanism for suppressing domain-specific information while maintaining a favorable balance between forgetting and retention.
>
> *Table D-4. Per-domain Accuracy on Office-Home.*
>
> ||art|clipart|product|real|
> |:---|:---:|:---:|:---:|:---:|
> |Zero-shot CLIP|74.34|60.97|80.43|81.29|
> |Ours (Forgotten = art)|**39.25**|70.18|88.72|75.91|
> |Ours (Forgotten = clipart)|77.36|**15.13**|87.87|80.00|
> |Ours (Forgotten = product)|80.38|72.15|**32.77**|77.85|
> |Ours (Forgotten = real)|68.30|67.98|78.94|**55.48**|
>
> ---
> **References**
>
> [a] Wu et al., Test-Time Domain Adaptation by Learning Domain-Aware Batch Normalization, AAAI 2024.
>
> [b] Cho et al., PromptStyler: Prompt-driven Style Generation for Source-free Domain Generalization, ICCV 2023.
>
> [c] Jhoo and Heo, Collaborative Learning with Disentangled Features for Zero-shot Domain Adaptation, ICCV 2021.
>
> [d] Mitsuzumi et al., Generalized Domain Adaptation, CVPR 2021.

---

> > ### Author Response · Authors · 2025-08-07
> >
> > Dear Reviewer ubkk,
> >
> > Thank you very much for your constructive comments and for acknowledging the main contributions of our work. We appreciate that you clearly recognized both the novelty of the Approximate Domain Unlearning (ADU) problem and the effectiveness of our proposed solution.
> >
> > We sincerely believe that our rebuttal has addressed all of the concerns you kindly raised.
> >
> > As there is not much time left in the reviewer-author discussion period, we just wanted to ask if there are any remaining points that may still be unclear. If not, we would greatly appreciate it if this could be reflected in your final review.
> >
> > Best regards,
> >
> > Authors of Paper ID 274

---

### Official Review · Reviewer_tNbB · 2025-07-03

**Clarity:** 3
**Significance:** 3
**Originality:** 3
**Rating:** 4
**Confidence:** 4

**Summary:**

This paper introduces Approximate Domain Unlearning (ADU), a new task that aims to make vision-language models forget specific domains (e.g., sketches) while retaining performance on others (e.g., real images). The authors propose a Domain Disentangling Loss and an Instance-wise Prompt Generator to address domain entanglement. Experiments on multiple benchmarks show clear improvements over existing unlearning and CLIP tuning baselines.

**Questions:**

- How does the method perform when domain definitions are noisy or ambiguous? For instance, can it still disentangle domains effectively if the boundary between “art” and “sketch” is unclear?

- Can the method selectively forget domain-specific features within a class without hurting class-level performance?

- It would be helpful to see how the Mem and For metrics evolve over training. Does forgetting conflict with memorization at certain stages, or are the gradients well-behaved?

- Is the method inherently tied to VLMs like CLIP, or could it work with plain vision encoders?

**Ethical Concerns:**

["NO or VERY MINOR ethics concerns only"]

**Final Justification:**

Thank you very much for the author’s response. It addressed all of my questions, so I’ll stick with my original score.

**Limitations:**

Yes

**Quality:**

2

**Strengths And Weaknesses:**

Strengths:

-  The paper introduces an interesting problem setting called Approximate Domain Unlearning. Instead of forgetting specific object classes, the goal here is to make the model forget certain domains like sketches or illustrations, which makes sense in scenarios where domain confusion can cause real issues.


- The method is interesting, it combines a domain disentangling loss to separate different domains in the feature space, and an Instance-wise Prompt Generator that lets the model adjust prompts based on each image, which helps deal with how mixed-up the domains usually are in VLMs.

- Well written, clear visualization.

Weaknesses

-  The method assumes you know the domain label for every training image, which might not always be the case. The paper mentions this, but doesn’t really give a backup plan if those labels aren’t available.

- The proposed DDL seems to work well, but it uses MMD, which is known to be sensitive to things like the choice of kernel and how batches are formed. The paper doesn’t really explain why their particular setup was chosen or how stable it is.

- The method only updates the vision prompts and keeps the rest of the model frozen. That makes it efficient, but also limits how much it can actually change the internal representations, especially if the domain-specific stuff is already mixed in early on. The paper doesn’t really talk about this trade-off or look into which layers are actually doing the forgetting.

- In all experiments, the domains to forget and retain are predefined and balanced in terms of class coverage. But in practice, the distribution of domains and classes is rarely so clean, (e.g., some domains might contain only a subset of classes or very different class frequencies). The paper doesn’t explore how the method handles such imbalanced or partially overlapping domain-class settings, which limits the understanding of its robustness.

---

> ### Author Rebuttal · Authors · 2025-07-27
>
> We sincerely thank Reviewer tNbB for their thoughtful and constructive review. We are especially grateful that the reviewer **recognized all the main contributions of our work**, including the introduction of Approximate Domain Unlearning (ADU) as a novel problem setting, the effectiveness of our proposed solution, and the clarity of our presentation. We believe that, as the first study to explore ADU, our paper can meaningfully contribute to the community. We also appreciate that the reviewer's comments were framed not as criticisms of fundamental flaws, but rather as constructive suggestions for further development. We view these points as valuable opportunities to deepen and expand our work, and we address them point-by-point below.
>
> ### **Q1. Domain labels not always fully available.**
> As we discuss in the Limitations section of our paper, we acknowledge that domain labels may not always be available for all training samples in real-world scenarios. However, it is important to note that most standard formulations in domain adaptation and domain generalization (e.g., [a–c]) assume full access to domain labels during training. Therefore, our setting aligns with this widely accepted convention and should not be considered an unrealistic simplification.
>
> Meanwhile, to evaluate the performance of our method under partial domain label availability, we consider a setting in which a portion of the training samples (ranging from 30% to 70%) lack domain annotations.
> To assess the effectiveness of combining domain estimation with our method in such cases, we adopt a simple pseudo-labeling approach: a domain classifier is trained using only the samples with known domain labels and then used to assign pseudo labels to the unlabeled samples.
>
> As shown in Table C-1, without domain estimation, the $H$-score drops from 75.89% to 49.97% under 70% missing domain labels. In contrast, with domain estimation, the $H$-score remains substantially higher at 61.21% even under the same missing rate. These results support our claim that our approach can be extended to more realistic cases where only partial domain annotations are available.
>
> *Table C-1. Performance on Office-Home with Partial Domain Labels.*
>
> | | |w/o Domain Estimation| | | w/ Domain Estimation| |
> |:---:|:---:|:---:|:---:|:---:|:---:|:---:|
> |Unlabeled sample ratio|$H\uparrow$|$Mem\uparrow$|$For\uparrow$|$H\uparrow$|$Mem\uparrow$|$For\uparrow$|
> |0|75.89|72.15|80.77|75.89|72.15|80.77|
> |0.3|**63.38**|83.35|52.68|**65.81**|83.06|55.98|
> |0.5|**55.77**|84.63|43.66|**63.64**|84.69|52.20|
> |0.7|**49.97**|83.84|36.98|**61.21**|83.37|50.00|
>
> ### **Q2. Choice of kernel and batch formation for MMD?**
> All results reported in the paper use the RBF (Gaussian) kernel. To assess sensitivity to kernel choice, we additionally tested four different kernels on Office-Home in Table C-2. Among them, the linear, Laplacian, and RBF kernels demonstrated generally stable performance, with only modest differences in accuracy. When validation-based kernel selection is feasible, we recommend choosing the best among these three; otherwise, the linear kernel serves as a conservative and reliable default.
>
> Regarding batch formation, we use a batch size of 128 and randomly select 128 samples per batch from the entire training data.
>
> *Table C-2. Impact of Kernel Choice.*
>
> | | |$\|D_{for}\|\=1$| | |$\|D_{for}\|\=2$| | |$\|D_{for}\|\=3$| |
> |:----|:----:|:----:|:----:|:----:|:----:|:----:|:----:|:----:|:----:|
> |Kernel|$H\uparrow$|$Mem\uparrow$|$For\uparrow$|$H\uparrow$|$Mem\uparrow$|$For\uparrow$|$H\uparrow$|$Mem\uparrow$|$For\uparrow$|
> |Linear|66.83|**79.85**|58.43|69.32|**77.35**|63.46|73.91|**74.02**|74.50|
> |Laplacian|68.50|79.00|61.44|71.18|75.93|67.17|73.66|71.69|76.54|
> |Polynomial|36.53|23.29|**84.68**|49.73|41.79|61.40|53.65|44.49|67.56|
> |RBF|**69.96**|77.93|64.34|**73.58**|75.61|**71.89**|**75.89**|72.15|**80.77**|
>
> ### **Q3. Concern about limited representation change in early layers due to model freezing.**
> Many prior studies have proven that fully fine-tuning CLIP often leads to catastrophic forgetting, especially in few-shot regimes [e, f]. Prompt tuning with frozen model parameters has thus become a standard alternative, which we follow. As shown in Table C-3, updating model parameters in our task indeed causes severe catastrophic forgetting, reinforcing this concern.
>
> Despite freezing, our method affects both low- and high-level representations. Fig. 7 shows stable unlearning performance across various prompt insertion depths, including early layers. This suggests that the learned prompts induce meaningful changes throughout the network, addressing the concern about limited adaptability in early stages.
>
> *Table C-3. Impact of Updating Model Parameters on Office-Home.*
>
> | | | $\|D_{for}\|\=1$ | | |$\|D_{for}\|\=2$| | | $\|D_{for}\|\=3$| |
> |:----|:----:|:----:|:----:|:----:|:----:|:----:|:----:|:----:|:----:|
> | Update Model Parameters?|$H\uparrow$|$Mem\uparrow$|$For\uparrow$|$H\uparrow$|$Mem\uparrow$|$For\uparrow$|$H\uparrow$|$Mem\uparrow$|$For\uparrow$|
> |Yes|3.90|2.00|**98.31**|2.95|1.50|**98.52**|3.64|1.86|**98.26**|
> |No (Ours)|**66.81**|**58.86**|77.23|**67.81**|**58.89**|79.90|**68.83**|**59.06**|82.48|
>
> ### **Q4. Robustness to Partial Domain-Class Overlap.**
> We conducted additional experiments on Office-Home where samples within three random classes (specifically, mug, radio, and hammer) were excluded from only "art" and "clipart" domains, creating a setting with partial domain-class overlap. As shown in Table C-4, even under such imbalance, our method maintains competitive $H$-scores. These results suggest that our method can handle realistic class-domain imbalances without critical loss in effectiveness.
>
> *Table C-4. Robstness to Partial Domain-Class Overlap.*
> | | | $\|D_{for}\|\=1$ | | | $\|D_{for}\|\=2$| | | $\|D_{for}\|\=3$| |
> |:----|:----:|:----:|:----:|:----:|:----:|:----:|:----:|:----:|:----:|
> ||$H\uparrow$|$Mem\uparrow$|$For\uparrow$|$H\uparrow$|$Mem\uparrow$|$For\uparrow$|$H\uparrow$|$Mem\uparrow$|$For\uparrow$|
> |Original Dataset|**69.96**|77.93|**64.34**|**73.58**|75.61|**71.89**|**75.89**|72.15|**80.77**|
> |Dataset w/ Partial Domain-Class Overlap |67.93|**81.81**|58.69|72.03|**79.61**|66.24|74.51|**75.12**|73.90|
>
> ### **Q5. Robustness to ambiguous domain definitions.**
> We agree that domains may overlap semantically or visually, making clear boundaries difficult. To examine this, we report per-domain accuracy on Office-Home in Tables C-5. Our method demonstrates selective forgetting and memorization even between highly similar domains, indicating effective disentanglement despite domain ambiguity.
>
> For robustness to label noise, our response to Q1 may offer a partial answer, as it addresses the case where domain labels are only partially available — a setting that can also be interpreted as involving random label noise on unlabeled samples (Table C-1).
>
> *Table C-5. Per-domain Accuracy on Office-Home.*
> ||art|clipart|product|real|
> |:---|:---:|:---:|:---:|:---:|
> |Zero-shot CLIP|74.34|60.97|80.43|81.29|
> |Ours (Forgotten = art)|**39.25**|70.18|88.72|75.91|
> |Ours (Forgotten = clipart)|77.36|**15.13**|87.87|80.00|
> |Ours (Forgotten = product)|80.38|72.15|**32.77**|77.85|
> |Ours (Forgotten = real)|68.30|67.98|78.94|**55.48**|
>
> ### **Q6. The method can forget domain knowledge without hurting class knowledge?**
> Yes, this is precisely the goal of ADU. The task is to forget domain-specific information (e.g., "sketch") while preserving class-relevant, domain-agnostic features. As shown in Table 1, our method significantly improves the forgetting score ($For$) on the target domain, while maintaining the memorization score ($Mem$) on the rest. These results demonstrate that the model can forget domain-specific representations of a class (e.g., "sketch" dog) without degrading its classification accuracy on the same class in other domains (e.g., "real" dog).
>
> ### **Q7. How $Mem$ and $For$ scores evolve over training?**
> As shown in Table C-6, we see that $For$ increases rapidly while $Mem$ drops in the early stages, suggesting a conflict between the two. However, both socres improve steadily after this phase, indicating that the model gradually learns to balance forgetting and retention. These results suggest a two-stage process: initial domain disentanglement followed by controlled forgetting and memorization.
>
> *Table C-6. $Mem$ and $For$ over Training Iterations on Office-Home.*
>
> |Iteration|0|20|40|60|80|100|200|500|800|
> |:----|:----|:----|:----|:----|:----|:----|:----|:----|:----|
> |$Mem\uparrow$|59.48|19.67|33.97|48.55|50.39|60.61|66.15|76.58|79.19|
> |$For \uparrow$ |48.06|85.61|67.56|64.27|64.84|63.41|64.48|84.77|85.36|
>
> ### **Q8. The method work with plain vision encoders?**
> Thank you for this valuable suggestion! Our primary focus is on pre-trained VLMs, particularly CLIP, where domain-specific representations are highly entangled. While our DDL and InstaPG are in principle applicable to plain vision encoders, the absence of public pre-trained vision-only models with domain generalization capabilities comparable to CLIP makes immediate evaluation difficult. We are interested in exploring this point in the final version.
>
> ---
> **References**
>
> [a] Wu et al., Test-Time Domain Adaptation by Learning Domain-Aware Batch Normalization, AAAI 2024.
>
> [b] Cho et al., PromptStyler: Prompt-driven Style Generation for Source-free Domain Generalization, ICCV 2023.
>
> [c] Jhoo and Heo, Collaborative Learning with Disentangled Features for Zero-shot Domain Adaptation, ICCV 2021.
>
> [d] Mitsuzumi et al., Generalized Domain Adaptation, CVPR 2021.
>
> [e] Zhou et al., Learning to Prompt for Vision-Language Models, IJCV 2022.
>
> [f] Kumar et al., Fine-tuning Can Distort Pretrained Features and Underperform Out-of-Distribution, ICLR 2022.

---

> > ### Comment · Reviewer_tNbB · 2025-08-05
> >
> > Thank you very much for the author’s response. It addressed all of my questions, so I’ll stick with my original score and lean toward accepting the paper.

---

> > > ### Author Response · Authors · 2025-08-05
> > >
> > > Thank you very much for your kind follow-up. We are pleased to hear that our response addressed all of your questions.
> > >
> > > As you kindly pointed out in your review, our work represents the first attempt at Approximate Domain Unlearning for Vision-Language Models. We are confident that this paper will provide novel insights and a new research direction to the machine learning community.
> > >
> > > If you have any further questions or requests, we would be more than happy to address them!

---

### Official Review · Reviewer_ZXGU · 2025-07-03

**Clarity:** 3
**Significance:** 3
**Originality:** 3
**Rating:** 4
**Confidence:** 3

**Summary:**

This paper introduces the problem of Approximate Domain Unlearning (ADU), which aims to reduce recognition accuracy for images from specified domains (e.g., clip art) while preserving accuracy for other domains (e.g., real images). Compared to existing approximate class unlearning tasks, ADU can be considered as a finer-grain task as it needs to distinguish different domains within the same class. With this finer-grain task, there are new challenges. For example, domain distributions are highly entangled in the feature space (e.g. real car images vs. illustrated car images) due to the strong domain generalization capability of pre-trained VLMs, which makes domain unclearning difficult. This paper proposes two approaches to tackle the challenges: Domain Disentangling Loss (DDL) and Instance-wise Prompt Generator (InstaPG).  Domain disentangling loss explicitly disentangles domain distributions in the latent space. Instance-wise Prompt Generator adaptively models instance-level domain variations.

Experiments and ablation studies are done on multi-domain image benchmark datasets, showing the effectiveness of the approach.

**Questions:**

1) In applications such as autonomous driving, data may be very unbalanced among different domains. For example, there would be a lot more real car images than illustrated car images (car on a poster).  Have you studied how sensitive domain disentangling loss is to such kind of data imbalance among domains?
2) As domain labels are not always readily available. How sensitive is your approach to noise in domain labels?
3) Have you tried other pre-trained VLMs?
4) Do you have any analysis on the computation cost, e.g., the increase with the use of InstaPG?

**Ethical Concerns:**

["NO or VERY MINOR ethics concerns only"]

**Final Justification:**

The paper introduces a new problem: Approximate Domain Unlearning (ADU). It also proposes two approaches to tackle the challenges in ADU. The proposed approach seems to be effective.

My questions were adequately answered in the rebuttal. Feedback from other reviewers is also generally positive. I would like to maintain my rating for acceptance.

**Limitations:**

Yes.

**Paper Formatting Concerns:**

No formatting concerns.

**Quality:**

3

**Strengths And Weaknesses:**

Strengths:
1) The paper proposes a new problem: Approximate Domain Unlearning (ADU).  This is useful for applications such as in self-driving, where car recognition would mean recognition of “real” cars on the road, instead of pictures of a car on a poster.
2) The paper proposes two methods to alleviate the entanglement between different domain distributions in the latent feature space: Domain Disentangling Loss (DDL) and Instance-wise Prompt Generator (InstaPG). DDL is based on the idea that if the feature distributions of individual domains are well-separated, the domain labels can be accurately predicted, and vice versa. InstaPG is to account for  instance-level variation in the images. The proposed approaches are intuitive and seem to be effective, as shown via experiments.
3) Experiments and ablation studies are done on multi-domain image benchmark datasets, including ImageNet, Office-Home, and Mini DomainNet. The experimental results demonstrated the effectiveness of the proposed approach.
4) The paper is generally well written.


Weakness:
1) One limitation of the proposed approach is that, for using the domain disentangling loss,  it assumes the availability of domain labels. As the authors pointed out, this may be a strong assumption, for example, domain labels are not readily available in autonomous driving data.
2) In applications such as autonomous driving, data may be very unbalanced among different domains. For example, there would be a lot more “real” car data compared to illustrated car data. It would be interesting to see how domain disentangling loss works in those situations.
3) The paper is based on a pre-trained CLIP model. It would be interesting to see results on other pre-trained VLMs.
4) The Instance-wise Prompt Generator (InstaPG) increases the training and inference latency. It would be nice to have some analysis on the computational cost.

---

> ### Author Rebuttal · Authors · 2025-07-28
>
> We sincerely thank Reviewer ZXGU for their thoughtful and insightful review. We are especially grateful that the reviewer **clearly recognized all the key contributions of our work**, including the novelty of the Approximate Domain Unlearning task, the effectiveness of our proposed methods (DDL and InstaPG), and the clarity and comprehensiveness of our experiments. We believe this paper makes the first concrete attempt at Approximate Domain Unlearning for Vision-Language Models, and we are confident that our contributions will meaningfully benefit the community by highlighting important challenges and proposing practical solutions in this emerging area. Below, we provide detailed responses to each comment and question. We greatly appreciate that all feedback was constructive and aimed at further strengthening the contribution, without pointing out any fundamental flaws.
>
> ### **Q1. Domain labels not always fully available.**
> As we discuss in the Limitations section of our paper, we acknowledge that domain labels may not always be available for all training samples in real-world scenarios. However, it is important to note that most standard formulations in domain adaptation and domain generalization (e.g., [a–c]) assume full access to domain labels during training. Therefore, our setting aligns with this widely accepted convention and should not be considered an unrealistic simplification.
>
> Meanwhile, to evaluate the performance of our method under partial domain label availability, we consider a setting in which a portion of the training samples (ranging from 30% to 70%) lack domain annotations.
> To assess the effectiveness of combining domain estimation with our method in such cases, we adopt a simple pseudo-labeling approach: a domain classifier is trained using only the samples with known domain labels and then used to assign pseudo labels to the unlabeled samples.
>
> As shown in Table B-1, without domain estimation, the $H$-score drops from 75.89% to 49.97% under 70% missing domain labels. In contrast, with domain estimation, the $H$-score remains substantially higher at 61.21% even under the same missing rate. These results support our claim that our approach can be extended to more realistic cases where only partial domain annotations are available.
>
> *Table B-1. Performance on Office-Home with Partial Domain Labels.*
>
> | | |w/o Domain Estimation| | | w/ Domain Estimation| |
> |:---:|:---:|:---:|:---:|:---:|:---:|:---:|
> |Unlabeled sample ratio|$H\uparrow$|$Mem\uparrow$|$For\uparrow$|$H\uparrow$|$Mem\uparrow$|$For\uparrow$|
> |0|75.89|72.15|80.77|75.89|72.15|80.77|
> |0.3|**63.38**|83.35|52.68|**65.81**|83.06|55.98|
> |0.5|**55.77**|84.63|43.66|**63.64**|84.69|52.20|
> |0.7|**49.97**|83.84|36.98|**61.21**|83.37|50.00|
>
> ### **Q2. Robustness of DDL to domain imbalance?**
> We agree that domain imbalance can arise in practical scenarios, where some domains may contain fewer samples than others (e.g., abundant "real" images vs. sparse "art" or "clipart" images in autonomous driving). To evaluate the robustness of our DDL under such imbalance, we conducted experiments on Office-Home by reducing the number of training samples from selected domains (i.e., "art" and "clipart"), while keeping the other domains fixed. Our default setting uses 8 shots, and we test the performance when this is reduced to 4 or 1 shot(s).
>
> As shown in Table B-2, our method maintains stable performance under domain imbalance. Even with just one sample from each of the "art" and "clipart" domains, the model achieves a $For$ of 73.85 and a $Mem$ of 64.31 for $|D_{for}| = 3$. These results indicate that our method can still perform effective domain disentangling and selective forgetting even in severely imbalanced scenarios.
>
> We appreciate this suggestion, as it offers an additional valuable perspective for further demonstrating the robustness and practical utility of our approach!
>
> *Table B-2. Robustness to Domain Imbalance.*
> | | | $\|D_{for}\|\=1$ | | | $\|D_{for}\|\=2$ | | | $\|D_{for}\|\=3$ | |
> |:----|:----:|:----:|:----:|:----:|:----:|:----:|:----:|:----:|:----:|
> |Number of shots for selected domains|$H\uparrow$|$Mem\uparrow$|$For\uparrow$|$H\uparrow$|$Mem\uparrow$|$For\uparrow$|$H\uparrow$|$Mem\uparrow$|$For\uparrow$|
> |8 shots|69.96|77.93|64.34|73.58|75.61|71.89|75.89|72.15|80.77|
> |4 shots|68.23|82.24|59.55|71.24|81.21|63.82|74.34|74.61|74.08|
> |1 shot|66.89|77.64|59.61|63.80|78.16|56.53|68.75|64.31|73.85|
>
>
> ### **Q3. Performance with other VLMs?**
>
> Although evaluating on diverse pre-trained VLMs would be ideal, such efforts are limited by the scarcity of publicly available CLIP-like pre-trained models. This would be a reason that most existing studies on CLIP fine-tuning or unlearning (e.g., [f–j]) focus solely on CLIP.
>
> Nevertheless, we report the performance of our method applied to another VLM, SigLIP [e], in Table B-3 using ImageNet. As shown, our method outperforms the baseline (the best competitor compared with our method in our main paper), indicating its strong effectiveness even if not coupled with CLIP. These results demonstrates that our approach is not specific to CLIP and can be extended to other VLMs.
>
> *Table B-3. Performance with SigLIP on ImageNet.*
> | |$H\uparrow$|$Mem\uparrow$|$For\uparrow$|
> |:----|:----:|:----:|:----:|
> |Zero-shot |46.47 |**63.26**| 36.73|
> |Baseline | 48.98| 60.32 |41.23|
> |Ours| **64.97**| 48.42 |**98.71**|
>
>
> ### **Q4. Computation complexity?**
> Table B-4 shows computation time of our method with and without InstaPG using NVIDIA RTX A4000 GPU. The results indicate that adding InstaPG introduces a very minor overhead, namely an additional 50 seconds for training and only 0.2 seconds for inference.
>
> *Table B-4. Training and Inference Times.*
> | |Training [s] |Inference [s] |
> |:----|:----|:----|
> |w/o InstaPG|501|13.1|
> |Ours|550|13.3|
>
> ---
> **References**
>
> [a] Wu et al., Test-Time Domain Adaptation by Learning Domain-Aware Batch Normalization, AAAI 2024.
>
> [b] Cho et al., PromptStyler: Prompt-driven Style Generation for Source-free Domain Generalization, ICCV 2023.
>
> [c] Jhoo and Heo, Collaborative Learning with Disentangled Features for Zero-shot Domain Adaptation, ICCV 2021.
>
> [d] Mitsuzumi et al., Generalized Domain Adaptation, CVPR 2021.
>
> [e] Zhai et al., Sigmoid Loss for Language Image Pre-Training, ICCV 2023.
>
> [f] Zhou et al., Learning to Prompt for Vision-Language Models, IJCV 2022.
>
> [g] Khattak et al., MaPLe: Multi-modal Prompt Learning, CVPR 2023.
>
> [h] Li et al., Vision-Language Model Fine-Tuning via Simple Parameter-Efficient Modification, EMNLP 2024.
>
> [i] Huang et al., LP++: A Surprisingly Strong Linear Probe for Few-Shot CLIP, CVPR 2024.
>
> [j] Kuwana et al., Black-box Forgetting, NeurIPS 2024.

---

> > ### Comment · Reviewer_ZXGU · 2025-08-05
> >
> > Thank you to the authors for the detailed rebuttal response. My concerns are all adequately addressed. I would like to maintain my rating for acceptance.

---

> > > ### Author Response · Authors · 2025-08-06
> > >
> > > Dear Reviewer ZXGU,
> > >
> > > We appreciate your kind feedback and are pleased that our response has clarified all of your concerns!
> > >
> > > As you have already acknowledged the main contributions of our work, we just wanted to ask if there are any remaining points that may still be unclear. If not, we would greatly appreciate it if this could be reflected in your final review.
> > >
> > > Best regards,
> > >
> > > Authors of Paper ID 274

---

### Official Review · Reviewer_gH1G · 2025-07-03

**Clarity:** 3
**Significance:** 2
**Originality:** 2
**Rating:** 5
**Confidence:** 4

**Summary:**

This paper introduces a new machine unlearning framework for Vision-Language Models. Unlike traditional class-level unlearning that removes specific object categories, this work selectively removes recognition capability for images from certain domains (e.g., illustrations, sketches) while preserving performance on other domains (e.g., real photos). The authors propose two key technical components: Domain Disentangling Loss to separate domain distributions in feature space, and Instance-wise Prompt Generator to handle within-domain style variations.

**Questions:**

Why not include additional metrics like domain classification accuracy after unlearning, or measures of knowledge retention for related but not-forgotten content?

**Ethical Concerns:**

["NO or VERY MINOR ethics concerns only"]

**Final Justification:**

Applying machine unlearning to domain generalization is well-motivated and novel. My major concerns like few-shot training examples and failure analysis have been adequately addressed.

**Limitations:**

Yes

**Quality:**

3

**Strengths And Weaknesses:**

Pros:

1. The problem is genuinely innovative and addresses practical limitations of existing unlearning approaches. The autonomous driving example (distinguishing real cars from illustrated advertisements) is compelling and highlights real-world relevance.

2. This work effectively tackles the core challenge of domain entanglement in VLMs by combining cross-entropy loss with MMD maximization. It intelligently handles intra-domain variability through attention-based prompt generation. The combination addresses both macro-level (domain separation) and micro-level (instance variation) challenges.

3. Extensive evaluation on three multi-domain datasets with different numbers of forgotten domains. Strong baselines adapted from state-of-the-art VLM fine-tuning and unlearning methods. And thorough ablation studies demonstrating the contribution of each component.

Cons:

1. Experiments use only 8 samples per domain for training, which seems quite small. Evaluation limited to relatively simple multi-domain datasets. Unclear how the approach scales to more complex domain distributions or larger datasets

2. The paper lacks discussion of scenarios where the method might fail or perform poorly. What happens when domain boundaries are ambiguous or when domains share significant visual similarity?

3. No detailed analysis of computational overhead, training time, or memory requirements compared to baseline methods.

---

> ### Author Rebuttal · Authors · 2025-07-27
>
> We sincerely thank Reviewer gH1G for their thoughtful and constructive review. We are especially grateful that the reviewer **clearly acknowledged all the main contributions of our paper**, including the novelty of the problem setting, the effectiveness of our proposed solution, and the comprehensiveness of our experiments. We believe that our paper makes the first attempt at approximate domain unlearning in Vision-Language Models and can meaningfully contribute to the community by addressing key challenges of this emerging task. Below, we provide point-by-point responses to all questions and concerns raised. All the comments are constructive and help strengthen our work. We also appreciate that none of the points raised indicate fundamental flaws in our core ideas or methodology.
>
> ### **Q1-1. 8 training samples per domain quite small.**
>
> Tuning of VLMs is typically performed under a few-shot protocol [a-d], and recent work on unlearning for VLMs [e] also adopts this standard setting. Accordingly, we follow the same protocol. As shown in Appendix A.1, we report results for varying numbers of samples (from 1 to 32), where our method consistently and significantly outperforms the baseline across all cases.
>
> ### **Q1-2. Results on more complex domain distributions or larger datasets?**
>
> We additionally evaluated our method on DomainNet, an established multi-domain benchmark dataset that comprises 177K test samples over six domains and 345 classes, making it one of the complex and large-scale datasets of its kind. The results are shown in Table A-1. Ours is clearly better than the baseline (the best competitor compared with our method in our main paper), which emphasizes the strong effectiveness of our method.
>
> *Table A-1. Results on DomainNet.*
>
> | | | $\|D_{for}\|\=1$ | | | $\|D_{for}\|\=2$ | | | $\|D_{for}\|\=3$ | |
> |:----|:----:|:----:|:----:|:----:|:----:|:----:|:----:|:----:|:----:|
> |Method|$H\uparrow$|$Mem\uparrow$|$For\uparrow$|$H\uparrow$|$Mem\uparrow$|$For\uparrow$|$H\uparrow$|$Mem\uparrow$|$For\uparrow$|
> |Zero-shot CLIP|36.85|53.28|28.16|36.85|53.28|28.16|36.85|53.28|28.16|
> |Baseline|38.45|55.69|29.36|39.36|55.81|30.40|38.79|55.26|29.88|
> |Ours|**66.81**|**58.86**|**77.23**|**67.81**|**58.89**|**79.90**|**68.83**|**59.06**|**82.48**|
>
> ### **Q2. Discussion on ambiguous or similar domains, with a note on failure cases.**
>
> To the best of our knowledge, there is currently no established multi-domain benchmark datasets having ambiguous domain boundaries. Instead, we report per-domain accuracy on Office-Home, Mini DomainNet, and DomainNet, those which include domains with substantial visual similarity (e.g., "sketch" vs. "quickdraw").
>
> The results are shown in Tables A-2 to A-4 below. Overall, they demonstrate that our method can selectively suppress classification accuracy on the forgotten domain while maintaining performance on the others, even those with similar visual properties. For instance, in Mini DomainNet, forgetting "sketch" reduces its accuracy significantly from 72.54% to 20.64%, whereas similar domains such as "clipart" and "painting" experience only minor drops (within 1%). These results indicate that our method provides fine-grained control even in the presence of overlapping domain features.
>
> Regarding a failure case, we found that unlearning can be more difficult for certain domains. For example, when forgetting "real" in Office-Home, our method decreases the accuracy from 81.29% to 55.48%, which, although significantly lower than the original zero-shot CLIP, still retains moderate recognition ability. One possible reason is that CLIP is extensively pre-trained on image-text pairs dominated by real-world photos, making the "real" domain more strongly encoded and harder to forget. Nonetheless, as shown in Table 1 of the main paper, our method consistently achieves strong forgetting performance on average across datasets. These results suggest that although our approach is generally effective, slight variations may occur depending on the VLM’s pretraining bias; as a potential remedy, one may consider incorporating techniques for mitigating dataset bias.
>
> *Table A-2. Per-domain Accuracy on Office-Home.*
>
> ||art|clipart|product|real|
> |:---|:---:|:---:|:---:|:---:|
> |Zero-shot CLIP|74.34|60.97|80.43|81.29|
> |Ours (Forgotten = art)|**39.25**|70.18|88.72|75.91|
> |Ours (Forgotten = clipart)|77.36|**15.13**|87.87|80.00|
> |Ours (Forgotten = product)|80.38|72.15|**32.77**|77.85|
> |Ours (Forgotten = real)|68.30|67.98|78.94|**55.48**|
>
> *Table A-3. Per-domain Accuracy on Mini DomainNet.*
>
> ||clipart|painting|real|sketch|
> |:---|:---:|:---:|:---:|:---:|
> |Zero-shot CLIP|80.64|78.10|87.94|72.54|
> |Ours (Forgotten = clipart)|**24.76**|75.40|83.97|73.97|
> |Ours (Forgotten = painting)|81.59|**30.32**|84.92|73.97|
> |Ours (Forgotten = real)|77.94|69.52|**32.06**|74.44|
> |Ours (Forgotten = sketch)|79.05|77.62|87.46|**20.64**|
>
> *Table A-4. Per-domain Accuracy on DomainNet.*
>
> ||clipart |infograph |painting |quickdraw |real |sketch|
> |:---|:---:|:---:|:---:|:---:|:---:|:---:|
> |Zero-shot CLIP|71.84|50.01|65.36|14.91|83.42|63.07|
> |Ours (Forgotten = clipart)|**33.37**|51.55|67.72|28.93|79.87|62.22|
> |Ours (Forgotten = infograph)|71.84|**13.05**|67.77|32.07|79.50|63.55|
> |Ours (Forgotten = painting)|73.59|52.21|**22.77**|29.64|79.86|63.21|
> |Ours (Forgotten = quickdraw)|73.60|51.33|67.99|**7.01**|81.79|63.79|
> |Ours (Forgotten = real)|71.61|50.52|64.40|29.56|**35.89**|64.05|
> |Ours (Forgotten = sketch)|70.81|51.62|67.20|29.62|81.26|**24.51**|
>
>
> ### **Q3. Computational complexity?**
>
> Table A-5 below summarizes GPU memory usage and training time with NVIDIA RTX A4000 GPU on Office-Home. While our method incurs only slightly higher computational cost than lightweight baselines such as CoOp and LP++, it remains on par with advanced CLIP fine-tuning methods such as MaPLe and CLIPFit. These results prove that our method is sufficiently efficient for practical use. Notably, adding InstaPG increases memory usage by only 1 GB and training time by just 1 minute, indicating minimal overhead.
>
> *Table A-5. Computational Complexity.*
>
> | |Memory [GB]|Time [s]|
> |:----|:----|:----|
> |CoOp [a]|1.9 | 283 |
> |MaPLe [b]| 10.4 | 539 |
> |BBF|2.5| 2882 |
> |CLIPFit| 11.0 | 340 |
> |LP++| 3.8 | 35 |
> |Ours w/o  InstaPG |9.7| 501 |
> |Ours |10.7| 550 |
>
> ### **Q4. Why not report domain classification accuracy?**
>
> According to the reviewer's suggestion, we reported domain classification accuracy before and after unlearning on Office-Home in Table A-6. Accuracy improved significantly from 25.80% before unlearning to 79.43% after unlearning when using our full method. Furthermore, when our two key components, i.e., DDL and InstaPG, are removed, the domain classification accuracy drops substantially to 31.06%, indicating that our method plays a critical role in achieving effective domain separation. We believe these results provide strong additional evidence of the effectiveness of our method, particularly in facilitating domain unlearning. We are grateful to the reviewer for directing our attention to this important perspective!
>
> *Table A-6. Domain Classification Accuracy When "Art" is Forgotten.*
>
> ||Domain Classification Accuracy|
> |:---|:---:|
> |Before Unlearning |25.80|
> |Ours w/o DDL and InstaPG|31.06|
> |Ours|**79.43**|
>
> ---------------------------------------------
> **References**
>
> [a] Zhou et al., Learning to Prompt for Vision-Language Models, IJCV 2022.
>
> [b] Khattak et al., MaPLe: Multi-modal Prompt Learning, CVPR 2023.
>
> [c] Li et al., Vision-Language Model Fine-Tuning via Simple Parameter-Efficient Modification, EMNLP 2024.
>
> [d] Huang et al., LP++: A Surprisingly Strong Linear Probe for Few-Shot CLIP, CVPR 2024.
>
> [e] Kuwana et al., Black-box Forgetting, NeurIPS 2024.

---

> > ### Author Response · Authors · 2025-08-07
> >
> > Dear Reviewer gH1G,
> >
> > Thank you very much for your constructive comments and for acknowledging the main contributions of our work. We appreciate that you clearly recognized both the novelty of the Approximate Domain Unlearning (ADU) problem and the effectiveness of our proposed solution.
> >
> > We sincerely believe that our rebuttal has addressed all of the concerns you kindly raised.
> >
> > As there is not much time left in the reviewer-author discussion period, we just wanted to ask if there are any remaining points that may still be unclear. If not, we would greatly appreciate it if this could be reflected in your final review.
> >
> > Best regards,
> >
> > Authors of Paper ID 274

---

> > ### Comment · Reviewer_gH1G · 2025-08-08
> >
> > Thank the authors for the clear and comprehensive rebuttal. All my concerns have been addressed. I will raise my score.

---

> > > ### Author Response · Authors · 2025-08-08
> > >
> > > Dear Reviewer gH1G,
> > >
> > > We are very glad that you found our response helpful. Although time is limited, we would be happy to provide further clarification if you have any additional questions.
> > >
> > > Best regards,
> > >
> > > Authors of Paper ID 274

---

### Note · Authors · 2025-08-12

Dear ACs and Reviewers,

We sincerely thank you for the time and thoughtfulness you devoted to reviewing our work. Your constructive and encouraging feedback is greatly appreciated. As we believe our rebuttal addressed all the concerns raised, we would like to take this opportunity to briefly summarize the core contributions of our work and how they were received.

### 1. Contributions of Our Paper
* **Novel Problem Setting**: While previous studies on machine unlearning for pre-trained vision-language models (VLMs) have focused on class-level unlearning, this paper is **the first to address domain-level unlearning** and introduces a novel problem setting called Approximate Domain Unlearning (ADU).
* **New Technical Challenge and Solution**: The strong domain generalization ability of pre-trained VLMs results in highly aligned latent spaces across domains. As a result, **straightforward applications of existing class-level unlearning methods to ADU fails to selectively forget specified domains**. We introduce two key techniques: (1) *Domain Disentangling Loss (DDL)*, which explicitly separates domain distributions in the latent space, and (2) *Instance-wise Prompt Generator (InstaPG)*, which captures instance-level domain variation.
* **Broader Impact**: By formulating this new task and proposing a dedicated solution, our work opens up new directions for unlearning research. It also contributes to broadening the applicability of pre-trained VLMs to practical domains where specific domain control is crucial.

### 2. Reviewers' Evaluation
* **Initial Evaluation**: **All four reviewers acknowledged the main contributions above as its strengths and recommended Borderline Accept**.
The weaknesses they pointed out did not identify any serious errors or flaws in our work, but rather suggested potential directions to further expand its applicability.
* **Rebuttal and Discussion Phase**: We responded to all points with additional experiments. **No further concerns were raised during the discussion, suggesting that the reviewers' questions were satisfactorily resolved**. We respectfully hope that the contributions of our paper, as well as our efforts during the discussion, will be fairly reflected in the final decision.

Once again, thank you for your thoughtful engagement. We hope our work contributes meaningfully to the community, particularly in the emerging area of knowledge control in pre-trained VLMs.

Best regards,

Authors of Paper ID 274

---

### Decision · Program_Chairs · 2025-09-17

**Decision:**

Accept (spotlight)

**Comment:**

This work proposes the new problem setting of Approximate *Domain* Unlearning (ADU). Where prior Unlearning works focused on removing knowledge of specific classes, ADU focuses on removing the ability of pretrained models to recognize images from a given domain (e.g., removing knowledge of clip art images) while still maintaining performance on images from remaining domains. The authors argue that ADU is even more difficult than traditional class unlearning, as VLM domain generalization capabilities lead domains to have overlapping representations in the model's feature space, while classes are typically more easily distinguished in the embedding spaces. To overcome this, the authors propose a Domain Disentangling Loss (DDL) to explicitly disentangle domains in the embedding space, and an Instance-wise Prompt Generator (InstaPG) component to adaptively model instance-level domain variations. While no other methods designed for the proposed ADU problem setting exist at this time, the authors develop reasonable baselines (e.g. finetuning, adapting a leading unlearning approach), which their method performs well against on evaluations over three multi-domain image benchmark datasets.

Reviewers initial evaluation was largely positive. Still, reviewers raised concerns regarding computational overhead [gH1G, ZXGU,  ubkk], the requirement of having domain labels [ZXGU,  tNbB,  ubkk], domain imbalance [ZXGU], experiments using only CLIP and no additional VLMs [ZXGU], and sensitivity to choice of kernel for MMD [tNbB]. However, the authors response to these concerns, including additional results, was satisfactory to the point where all active reviewers have acknowledged that their concerns have been adequately addressed.

The proposed problem setting of ADU is novel while being well-justified, and one can imagine the utility and necessity of a solution to this problem, indicating that this is a worthy line of research. The technical challenges of ADU, specifically domains not being easily separated in VLM latent spaces, are interesting and the proposed approach is novel and sound. The authors' rebuttal was comprehensive, and the new results strengthen the already good work. For these reasons, I recommend acceptance.

Furthermore, I recommend a spotlight for this paper. The novel problem setting introduced in this work is, in my opinion, an interesting and important task. Highlighting this newly defined task setting would be valuable for the research community. The proposed approach for this new task is likewise novel and we'll justified. While some reviewers still had reservations about the need for domain labels, I do not believe this is a critical flaw. The need for domain labels is standard for most domain shift approaches, so I don't believe this is too strong of an assumption, and the semi-supervised results strengthen the work. So while it's a limitation, I don't think it's too strong of one.